# Modular DNA barcoding of nanobodies enables multiplexed in situ protein imaging and high-throughput biomolecule detection

**Shilin Zhong[1,2,3], Ruiyu Wang[1,2,3,4], Xinwei Gao[4], Qingchun Guo[4,5], Rui Lin[2,6]\*, Minmin Luo[4,7,8,9]\***

[1]School of Life Sciences, Peking University, Beijing, China; [2]National Institute of Biological Sciences (NIBS), Beijing, China; [3]Peking University-Tsinghua University-National Institute of Biological Sciences Joint Graduate Program, NIBS, Beijing, China; [4]Chinese Institute for Brain Research, Beijing, China; [5]School of Biomedical Engineering, Capital Medical University, Beijing, China; [6]Tsinghua Institute of Multidisciplinary Biomedical Research (TIMBR), Beijing, China; [7]New Cornerstone Science Laboratory, Shenzhen, China; [8]Research Unit of Medical Neurobiology, Chinese Academy of Medical Sciences, Beijing, China; [9]Beijing Institute for Brain Research, Chinese Academy of Medical Sciences & Peking Union Medical College, Beijing, China

**\*For correspondence:**
linrui@nibs.ac.cn (RL);
luominmin@cibr.ac.cn (ML)

## eLife Assessment

This **fundamental** manuscript presents a practical modification of the orthogonal hybridization chain reaction (HCR) technique, a promising yet underutilized method with broad potential for future applications across various fields. The authors advance this technique by integrating peptide ligation technology and nanobody-based antibody mimetics-cost-effective and scalable alternatives to conventional antibodies-into a DNA-immunoassay framework that merges oligonucleotide-based detection with immunoassay methodologies. Notably, they demonstrate with **compelling** evidence that this approach facilitates a modified ELISA platform capable of simultaneously quantifying multiple target protein expression levels within a single protein mixture sample.

**Abstract** Current immunodetection methods using antibody-DNA conjugates enable multiplexed target detection through orthogonal DNA barcodes, but existing conjugation approaches are labor-intensive and often compromise antibody function. Here, we present a modular, site-specific, and cost-efficient DNA tagging strategy – multiplexed and modular barcoding of antibodies (MaMBA). Utilizing nanobodies as modular adaptors, MaMBA enables direct site-specific labeling of off-the-shelf IgG antibodies with a one-component design. We first applied MaMBA to develop the MaMBA-assisted immunosignal hybridization chain reaction (*mis*HCR) method for highly multiplexed in situ protein imaging via orthogonal HCR. Its cleavable variant, *mis*HCRⁿ, achieves simultaneous visualization of 12 different targets within the same mouse brain sections through iterative probe use. We further extended the cleavable MaMBA to develop the barcode-linked immunosorbent assay (BLISA) for multiplexed and high-throughput biomolecule detections. By combining BLISA with next-generation sequencing, we successfully measured SARS-CoV-2 IgG and hepatitis B virus (HBV)-associated antigens in a large number of human serum samples. Additionally, we demonstrated a small-scale drug screen by using BLISA to simultaneously detect eight protein targets. In conclusion,

MaMBA offers a highly modular and easily adaptable approach for antibody DNA barcoding, which can be broadly applied in basic research and clinical diagnostics.

## Introduction

Accurate detection of target molecules in biological samples is key for biomedical research and clinical diagnostics. Owing to their ease of use, speed, and accessibility, immunodetections are currently the standard and most popular methods for biomolecule detection (*Maynard and Georgiou, 2000*; *Scott et al., 2012*). These methods use a primary antibody that binds selectively to a target molecule (antigen). This primary antibody-antigen interaction can be visualized via a conjugated reporter (e.g., horseradish peroxidase [HRP] or fluorophore) on the antibody. In most cases, a labeled secondary antibody, which can recognize the primary antibodies from the same host species, reacts with the antibody-antigen complex to generate readout signals. In classic immunodetection methods, chromogen, fluorescence, or luminescence is utilized to optically report the abundance and localization of target molecules. However, the overlaps of antibody species and reporter spectrums limit the simultaneous detection of only a few (often up to three) target molecules in a sample (*Stack et al., 2014*).

DNA barcoding represents a promising strategy to improve the multiplexity and efficiency of conventional immunodetection methods. This technique involves conjugating antibodies with DNA molecules containing unique barcode sequences, rather than traditional optical reporters. By utilizing a combination of DNA-labeled antibodies, multiple target molecules can be simultaneously detected within a single sample (*Lewis et al., 2021*; *Rajagopalan et al., 2021*). Each antibody-antigen interaction can be examined via distinct barcode sequences. Considering the high diversity of DNA sequences, DNA barcoding offers much higher multiplexity far beyond the scope of conventional reporter molecules. Importantly, careful design of DNA barcode sequences also enables pooled analyses using advanced DNA sequencing techniques, leveraging massively parallel sequencing capabilities to dramatically increase the detection throughput at low cost (*van Buggenum et al., 2018*; *Liszczak and Muir, 2019*; *Zhong et al., 2021*). However, the process of conjugating DNA oligonucleotides to individual primary antibodies is both labor-intensive and expensive, which severely limits the scalability of DNA barcoding. Moreover, existing DNA conjugation methods often employ non-site-specific approaches that target active functional groups on a limited number of amino acids (e.g., lysine and cysteine). This can result in reduced signals and decreased antibody affinity and specificity (*Dovgan et al., 2019*; *Dugal-Tessier et al., 2021*). Overcoming these obstacles is crucial for realizing the full potential of DNA barcoding in immunodetection applications.

Here, we present the multiplexed and modular barcoding of antibodies (MaMBA) strategy, a simple and cost-effective method of barcoding antibodies with single-stranded DNAs. To enhance modularity, MaMBA introduces nanobodies as adaptor proteins between the IgG antibodies and the barcode DNA oligos. Nanobodies are small (~15 kDa vs. ~50 kDa for protein A, protein G, or the Fab fragments) and can be recombinantly expressed in large quantities. This modular design enables the rapid dissemination of the DNA barcoding method to a vast number of off-the-shelf antibodies. MaMBA employs an enzymatic reaction for site-specific conjugation of DNA oligos to nanobodies, rather than using chemical linkers, thereby minimizing potential adverse effects on nanobody affinity and specificity. We first combined MaMBA with the fluorescent signal amplification method isHCR (immunosignal hybridization chain reaction) to establish a highly multiplexed and sensitive immunohistochemistry method for spatial protein profiling (MaMBA-assisted isHCR [*mis*HCR]). To further increase the multiplex capacity of *mis*HCR, we designed a cleavable MaMBA variant, enabling the development of a multi-round version of *mis*HCR (*mis*HCR$^n$). Extending this strategy to the traditional enzyme-linked immunosorbent assay (ELISA), we developed a barcode-linked immunosorbent assay (BLISA) and demonstrated its utility in multiplexed and high-throughput detections of disease-associated antigens, antibodies, and phosphoproteins.

## Results

### Design and validation of MaMBA

We used an enzymatic reaction to efficiently conjugate nanobodies with DNA oligos in a site-specific manner (*Figure 1a*). The recombinant *Oldenlandia affinis* asparaginyl endopeptidase (OaAEP1; *Yang*

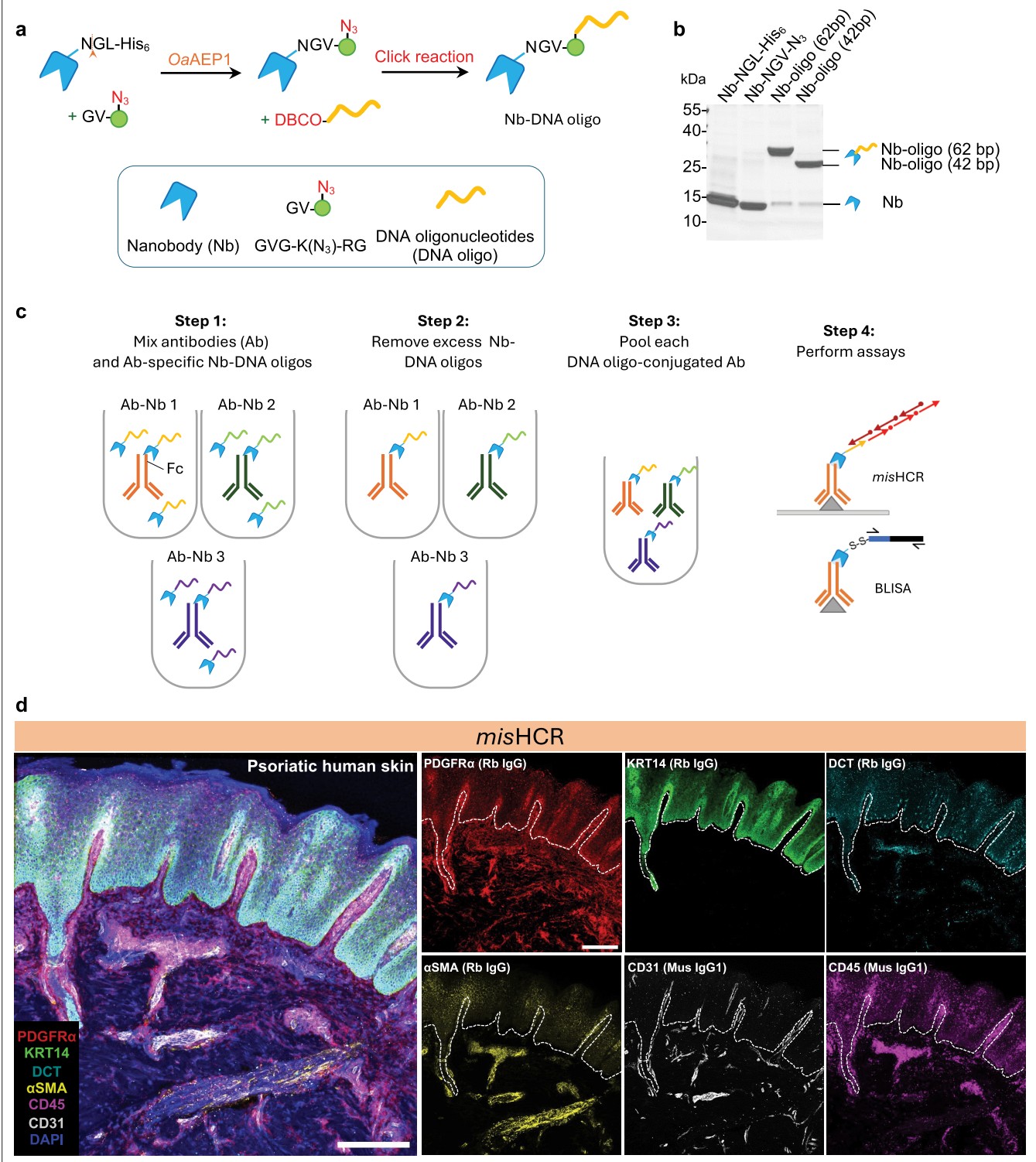

**Figure 1.** Development and application of multiplexed and modular barcoding of antibodies (MaMBA) and MaMBA-assisted immunosignal hybridization chain reaction (*mis*HCR). (**a**) Schematics of the DNA oligo-conjugated nanobody production process. (**b**) Coomassie-stained SDS-PAGE gel of purified nanobody (Nb-NGL-His$_6$), OaAEP1 reaction product (Nb-NGV-N$_3$), and click reaction products (Nb-DNA oligo). (**c**) MaMBA workflow. IgG antibodies are separately incubated with DNA oligo-conjugated nanobodies bearing unique DNA barcodes, forming antibody-Nb-DNA oligo complexes (Ab-Nb 1 to Ab-Nb 3). These complexes are subsequently purified using ultracentrifugal filters or IgG-conjugated beads, pooled, and applied to various assays. (**d**) Fluorescence confocal images of human psoriatic skin section stained for DAPI and six antigens by three rounds of

*Figure 1 continued on next page*

*Figure 1 continued*

*mis*HCR. IgG types of the employed antibodies are indicated in parentheses. Scale bar, 300 μm. Rb, rabbit; Mus, mouse; Th, tyrosine hydroxylase; PDGFRα, platelet-derived growth factor receptor A; KRT14, keratin 14; DCT, dopachrome tautomerase; αSMA, alpha-smooth muscle actin.

The online version of this article includes the following source data and figure supplement(s) for figure 1:

**Source data 1.** Original file for PAGE gel displayed in *Figure 1b*.

**Source data 2.** PNG file for PAGE gel in *Figure 1b*, indicating the relevant bands.

**Source data 3.** Source image file in *Figure 1d*.

**Figure supplement 1.** Efficiency assessment of nanobody-DNA oligo conjugation.

**Figure supplement 1—source data 1.** Original file for PAGE gel displayed in *Figure 1—figure supplement 1a*.

**Figure supplement 1—source data 2.** PNG file for PAGE gel in *Figure 1—figure supplement 1a*, indicating the relevant bands.

**Figure supplement 2.** Comparison of antibody-DNA oligo labeling methods.

**Figure supplement 2—source data 1.** Original files for gels and western blots displayed in *Figure 1—figure supplement 2b–e*.

**Figure supplement 2—source data 2.** PNG files for gels and western blots in *Figure 1—figure supplement 2b–e*, indicating the relevant bands.

**Figure supplement 3.** Characterization of multiplexed and modular barcoding of antibodies-assisted immunosignal hybridization chain reaction (*mis*HCR).

**Figure supplement 3—source data 1.** Source image files in *Figure 1—figure supplement 3b*.

**Figure supplement 3—source data 2.** Source image files in *Figure 1—figure supplement 3e*.

**Figure supplement 4.** Efficient and reproducible hybridization chain reaction (HCR) signal removal by formamide treatment.

**Figure supplement 4—source data 1.** Source image files in *Figure 1—figure supplement 4b*.

**Figure supplement 4—source data 2.** Source image files in *Figure 1—figure supplement 4c*.

**Figure supplement 5.** Multiplexed and modular barcoding of antibodies-assisted immunosignal hybridization chain reaction (*mis*HCR) imaging of human psoriatic and healthy skin sections.

**Figure supplement 5—source data 1.** Source image files in *Figure 1—figure supplement 5*.

**Figure supplement 6.** Comparison of the amine-based, thiol-based, and multiplexed and modular barcoding of antibodies (MaMBA)-based antibody labeling methods.

*et al., 2017*; *Rehm et al., 2019*) is employed to catalyze the ligation of an azide-bearing dipeptide substrate (Gly-Val, GV) to the nanobodies at the tripeptide recognition motif (Asn-Gly-Leu, NGL) fused at the C-termini. The resulting azide-functionalized nanobodies can subsequently be covalently tagged with DNA oligos of desired sequences and lengths via a single-step click reaction (*Figure 1a and b*). We chose nanobodies that selectively and monovalently bind to the fragment crystallizable (Fc) region of the IgG (*Pleiner et al., 2018*), thereby minimizing the potential interference with antigen recognition by the IgG. This property enables modular barcoding of antibodies through the straight-forward combination of off-the-shelf primary IgG antibodies with DNA oligo-conjugated nanobodies (Nb-DNA oligos) (*Figure 1c*). For the initial proof-of-principle demonstrations, we tested two IgG secondary nanobodies: TP897 (for rabbit IgGs) and TP1107 (specific for mouse IgGs; *Pleiner et al., 2018*). After tagging each IgG antibody with a unique DNA barcode, multiple antibodies can then be pooled together for subsequent applications (*Figure 1c*).

To assess the efficiency of the OaAEP1-mediated reaction for adding azide functional group to the nanobody and the efficiency of the click reaction for nanobody-DNA oligo conjugation, we performed SDS-PAGE gel electrophoresis of the protein products from each step (*Figure 1—figure supplement 1*). We observed high efficiency for both conjugation steps: 86.8% for the OaAEP1-mediated reaction (*Figure 1—figure supplement 1a and b*) and 92.9% for nanobody-DNA oligo conjugation (*Figure 1—figure supplement 1c and d*). We further measured the labeling degree of DNA molecules on IgG by MaMBA. We barcoded an anti-GFP IgG antibody with 62 bp DNA oligos using MaMBA alongside two classic conjugation methods (*Figure 1—figure supplement 2*). For both amine- and thiol-targeted conjugation, increasing the DNA:antibody ratio in the reactions enhanced conjugation efficiency but resulted in substantially higher labeling degree, which refers to the number of labels attached per anti-body molecule (*Figure 1—figure supplement 2a–c*). In stark contrast, MaMBA consistently labeled the antibodies with the same degree (one to two DNA molecules per IgG), irrespective of the reaction conditions (*Figure 1—figure supplement 2d*). We further examined the effects of DNA conjugation on the antibody's binding ability. We performed a western blot to detect GFP in cell lysates using

DNA-labeled IgG antibodies. Under identical DNA-to-antibody ratios (3:1 or 15:1) in the conjugation reactions, antibodies generated by MaMBA exhibited superior GFP detection compared to those generated by amine- or thiol-targeted methods (*Figure 1—figure supplement 2e*). The sensitivity of amine- or thiol-conjugated antibodies in detecting GFP deteriorated dramatically as the DNA-to-antibody ratio (3:1, 15:1, and 25:1) increased in the conjugation reactions (i.e., antibodies have higher labeling degree; *Figure 1—figure supplement 2e*). These results demonstrate the high efficiency and accuracy of MaMBA for antibody DNA barcoding.

## Multiplexed in situ biomolecule imaging by *mis*HCR

We evaluated the capability of MaMBA for multiplexed in situ biomolecule imaging using immunofluorescence techniques. In the original isHCR protocol that we developed for amplifying immunosignals (*Lin et al., 2018*), we conjugated HCR initiators to secondary antibodies via a streptavidin-biotin bridge. While effective, this approach necessitates four mandatory steps prior to HCRs (*Figure 1—figure supplement 3a*) and faces multiplexity limitations due to the use of streptavidin. To address these constraints, we used MaMBA to tag primary antibodies with HCR initiator-conjugated nanobodies. The resulting HCR initiator-primary antibody complexes (Ab-HCR initiator) can be directly applied to tissue samples, followed by HCR amplification (*Figure 1—figure supplement 3a*). MaMBA allows for the large-scale preparation and long-term storage of these key reagents, including the nanobody-HCR initiator conjugation and the MaMBA-barcoded primary antibodies, thereby effectively simplifying the isHCR procedure to a single immunostaining step. We named this new protocol *mis*HCR. To examine *mis*HCR's performance in tissue samples, we immunostained mouse brain sections against two target antigens (e.g., neuronal nuclei [NeuN] or tyrosine hydroxylase [Th]) using *mis*HCR or antibodies tagged with fluorophore-conjugated nanobodies (*Figure 1—figure supplement 3b*; see Materials and methods). *mis*HCR drastically increased the fluorescence signals (*Figure 1—figure supplement 3b–d*) and preserved spatial resolution for diffraction-limited confocal imaging (*Figure 1—figure supplement 3e–g*).

We next asked whether *mis*HCR has a high multiplexity capacity (*Figure 1—figure supplement 4a*). By barcoding primary antibodies with orthogonal HCR initiators, *mis*HCR resolves the constraints imposed by antibody host species and enables simultaneous immunostaining of multiple targets in the same sample. Visualization of antigen-antibody-HCR initiator complexes is achieved through orthogonal HCR amplification reactions. To resolve the issue of spectral overlaps among fluorescent reporters, we established a protocol for the efficient removal of fluorescence signals from previous HCR amplifications through formamide-mediated dehybridization of HCR amplifiers (*Figure 1—figure supplement 4b*; see Materials and methods). We validated this approach by conducting five consecutive cycles of HCR amplification/formamide treatment in immunostaining mouse brain sections against NeuN and neurofilament H (NF-H) (*Figure 1—figure supplement 4c*). Statistical analysis revealed no significant differences in fluorescence intensity between consecutive rounds (*Figure 1—figure supplement 4d*), with over 80% of fluorescence signals retained after four rounds of formamide treatment (*Figure 1—figure supplement 4d*). Thus, formamide did not significantly disrupt the antigen-Ab-HCR initiator complexes nor impair the subsequent HCR amplification reactions, allowing for sequential rounds of HCR amplification and thus highly reproducible immunofluorescence detection.

As a demonstration of its potential, we applied *mis*HCR for multiplexed immunostaining on skin biopsies from healthy donors or patients with psoriasis (*Figure 1d*). We used four rabbit IgG antibodies and two mouse IgG1 antibodies to detect markers of six cell types. *Mis*HCR successfully revealed distinct patterns of each cell type (*Figure 1d*). Consistent with previous findings (*Lowes et al., 2007*; *Bowcock and Krueger, 2005*), the skin biopsies from psoriasis patients exhibited dilated blood vessels, accumulation of white blood cells, and abnormal proliferation of keratinocytes (*Figure 1d* and *Figure 1—figure supplement 5*).

To further enhance the multiplexing capability of *mis*HCR, we developed a cleavable variant of MaMBA which incorporates an additional disulfide linker between the nanobody and the conjugated DNA oligo (*Figure 2a*). Upon exposure to reducing agents such as tris(2-carboxyethyl)phosphine (TCEP), the disulfide linker undergoes reductive cleavage, releasing the conjugated DNA oligo from the nanobody. By integrating this cleavable MaMBA tagging with *mis*HCR, all HCR initiators (and their associated hybridized HCR amplifiers) can be removed from the samples post-imaging (*Figure 2—figure supplement 1*; see Materials and methods). This approach enables the reuse of the same set

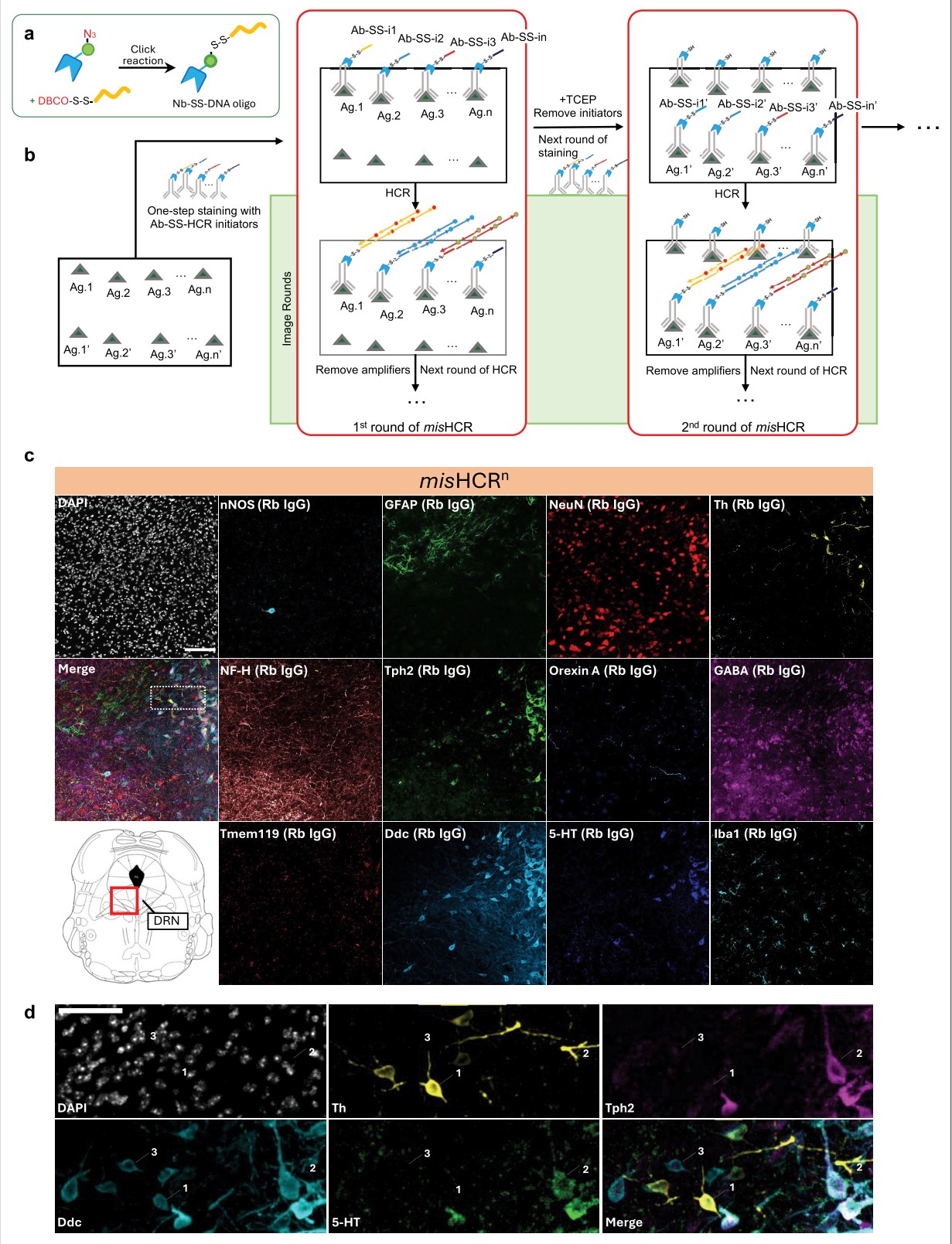

**Figure 2.** Multi-round immunostaining using *mis*HCRⁿ. (**a**) Schematics of the production process for cleavable DNA oligo-conjugated nanobodies. (**b**) Workflow of *mis*HCRⁿ staining and imaging. The process begins with the first round of staining (1st round of *mis*HCR), where a pool of primary antibodies barcoded with orthogonal cleavable hybridization chain reaction (HCR) initiators (Ab-SS-i1 to Ab-SS-in) is applied to the sample simultaneously. This is followed by sequential rounds of HCR imaging (image rounds). Upon completion of the image rounds, HCR initiators are

*Figure 2 continued on next page*

*Figure 2 continued*

removed using TCEP to cleave the disulfide bonds, enabling the start of the second round of staining (2nd round of *mis*HCR). Antibodies used in the 2nd round of *mis*HCR can be equipped with the same set of HCR initiators as in the 1st round of *mis*HCR (Ab-SS-i1' to Ab-SS-in'). This process can be iteratively performed for multi-round *mis*HCR. (**c**) Fluorescence confocal images of the dorsal raphe nucleus (DRN) in a mouse brain section, stained for 12 antigens by *mis*HCR (*Scott et al., 2012*) and counterstained with DAPI. The imaging location is highlighted in the red boxed region of the mouse brain atlas shown at the bottom left. IgG types of the employed antibodies are indicated in parentheses. Scale bar, 100 μm. (**d**) Zoomed-in views of the region marked by a dotted box in panel (**c**). Arrows indicate three cell types (1, $Ddc^+Th^+Tph2^-$ cells; 2, $Ddc^+Th^-Tph2^+$ cells; 3, $Ddc^+Th^-Tph2^-5-HT^-$ cells). Scale bar, 50 μm. nNOS, neuronal nitric oxide synthase; GFAP, glial fibrillary acidic protein; NF-H, neurofilament H; Tph2, tryptophan hydroxylase 2; GABA, gamma-aminobutyric acid; Tmem119, transmembrane protein 119; Ddc, DOPA decarboxylase; 5-HT, 5-hydroxytryptamine; Iba1, ionized calcium binding adaptor molecule.

The online version of this article includes the following source data and figure supplement(s) for figure 2:

**Source data 1.** Source image file in *Figure 2c*.

**Figure supplement 1.** Efficient multiplexed and modular barcoding of antibodies-assisted immunosignal hybridization chain reaction (*mis*HCR) signal removal by TECP treatment.

**Figure supplement 1—source data 1.** Source image files in *Figure 2—figure supplement 1*.

**Figure supplement 2.** *mis*HCR$^n$ for immunostaining nine target antigens using three orthogonal hybridization chain reaction (HCR) initiators.

**Figure supplement 2—source data 1.** Source image file in *Figure 2—figure supplement 2*.

**Figure supplement 3.** Validation of the specificity of multiplexed and modular barcoding of antibodies-assisted immunosignal hybridization chain reaction (*mis*HCR).

**Figure supplement 3—source data 1.** Source image files in *Figure 2—figure supplement 3*.

of HCR initiators with additional antibodies in subsequent rounds of immunostaining. Through iterative cycles of multiplexed immunostaining, sequential HCR amplification, imaging, and HCR initiator removal, we achieve highly flexible and multiplexed in situ biomolecule detection (*Figure 2b*). We named this new pipeline *mis*HCR$^n$.

We validated *mis*HCR$^n$ by immunostaining mouse brain sections against nine target antigens with well-defined and distinct distributions (*Figure 2—figure supplement 2a*). We used nine rabbit IgG antibodies and three orthogonal HCR initiators to perform three rounds of *mis*HCR (*mis*HCR$^3$). All targets were successfully visualized (*Figure 2—figure supplement 2a*). Protein markers for cell types showed distinct and nonoverlapping distributions (*Figure 2—figure supplement 2b*). No cross-reaction among barcoded antibodies from the same or different rounds of *mis*HCR was observed. We then further demonstrated a 12-antigen in situ imaging in the dorsal raphe nucleus (DRN) of mouse brain sections by two rounds of *mis*HCR (*mis*HCR$^2$) using 12 rabbit IgG antibodies with nine orthogonal HCR initiators (*Figure 2c*). Additional control immunostaining experiments confirmed the specificity of *mis*HCR on all 12 targets (*Figure 2—figure supplement 3*; see Materials and methods). We again observed precise distributions of target antigens. In particular, by analyzing the co-localization among DOPA decarboxylase (Ddc), Th, tryptophan hydroxylase 2 (Tph2), and 5-hydroxytryptamine (5-HT), we were able to identify three types of Ddc-expressing neurons in the DRN: likely dopamine neurons ($Ddc^+Th^+Tph2^-$), likely 5-HT neurons ($Ddc^+Th^-Tph2^+$), and the non-dopamine and non-5-HT neurons ($Ddc^+Th^-Tph2^-5-HT^-$; *Ikemoto, 2017*; *Figure 2d*). These results demonstrate that *mis*HCR utilizing MaMBA for antibody barcoding allows specific, flexible, and highly multiplexed in situ biomolecule imaging of animal and human clinical samples.

## Barcode-linked immunosorbent assay

Building on the success of MaMBA-based immunofluorescent imaging, we next explored the potential of MaMBA for high-throughput, quantitative assays of diverse biomolecules. The ELISA has long been a cornerstone technique in diagnostics and research for quantitative biomolecule analysis. Standard ELISAs predominantly rely on enzymatic reactions that utilize HRP or alkaline phosphatase to convert chromogenic substrates for sensitive signal readout. However, the limited availability of orthogonal enzyme-substrate pairs typically necessitates separate ELISA reactions for detecting multiple antigens within a single sample. To enhance multiplexing capacity, we integrate the cleavable version of MaMBA into the classic ELISA. We label antibodies targeting different antigens with DNA oligos harboring orthogonal barcode sequences (DNA barcodes), so the labeled antibodies can be applied to the sample simultaneously. Following the removal of excessive antibodies that do not bind

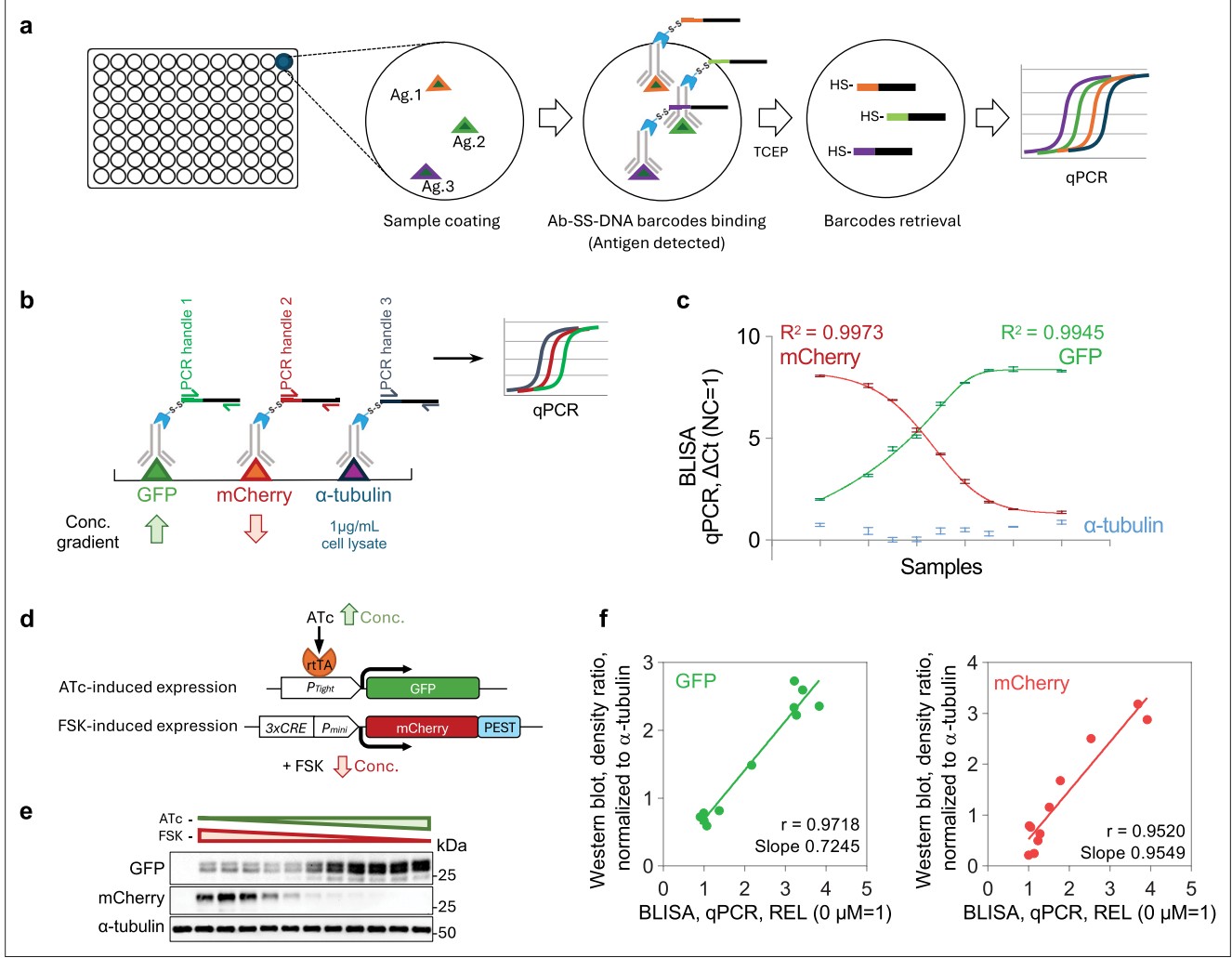

**Figure 3.** Barcode-linked immunosorbent assay (BLISA). (**a**) Workflow of direct antigen detection using BLISA. (**b**) Schematics of multiplexed detection for GFP, mCherry, and α-tubulin using BLISA. (**c**) Quantification of spiked-in purified GFP, spiked-in purified mCherry, and endogenous α-tubulin in cell lysates using quantitative PCR (qPCR)-based BLISA. Negative control (NC) represents cell lysates without spiked-in proteins. ΔCt values are normalized to the NC group, where ΔCt (NC = 1) = $Ct_{NC} - Ct_{Target} + 1$. The dose-response curves for purified GFP (green) and mCherry (red) are fitted using a five-parameter logistic (5PL) function. n=3 replicates; mean ± s.e.m. (**d**) Schematics of two drug-inducible gene expression systems for expressing GFP and mCherry in HEK293T cells. (**e**) Western blot of cell samples treated with varying concentrations of ATc and FSK. (**f**) Pearson's correlation between BLISA and western blot results for measuring relative expression levels (REL, normalized to α-tubulin) of GFP (left) and mCherry (right). REL = $2^{-(CtGFP/mCherry-Ct\alpha-tubulin)}$. ATc, anhydrotetracycline hydrochloride; FSK, forskolin.

The online version of this article includes the following source data and figure supplement(s) for figure 3:

**Source data 1.** Original files for western blots displayed in *Figure 3e*.

**Source data 2.** PDF files for western blots in *Figure 3e*, indicating the relevant bands.

**Figure supplement 1.** Barcode-linked immunosorbent assay (BLISA) for detecting proteins endogenously expressed in cells.

**Figure supplement 1—source data 1.** Original files for western blots displayed in *Figure 3—figure supplement 1b and e*.

**Figure supplement 1—source data 2.** PDF files for western blots in *Figure 3—figure supplement 1b and e*, indicating the relevant bands.

antigens, the DNA oligos are retrieved via the reductive cleavage for readout, which reduces potential interference from other DNAs (e.g., genome DNA). The quantity of target molecules in the samples can then be inferred by quantifying the corresponding DNA barcodes. We named this multiplexed biomolecule detection protocol BLISA.

We first established the BLISA protocol by detecting antigens that are directly immobilized on microplate surfaces (*Figure 3a*). We spiked recombinant GFP and mCherry proteins into cell lysates at varying concentrations. Each sample contained an equal amount of cell lysate, with the concentrations

of GFP and mCherry adjusted to create a gradient (*Figure 3b*). We simultaneously detected GFP, mCherry, and α-tubulin (an endogenous protein in the cell lysates) by a pool of three cleavable DNA-barcoded primary antibodies (Ab-SS-DNA barcodes), each specific to one antigen and tagged with a unique DNA oligo containing a distinct PCR handle sequence. The DNA barcodes were released by the reductant (i.e., TCEP) and quantified via orthogonal quantitative PCR (qPCR) (*Figure 3a and b*). Three targets were successfully detected by qPCR in all sample groups (*Figure 3c*). The ΔCt of qPCRs detecting GFP and mCherry in each group was well fitted with the predetermined concentration gradients of GFP and mCherry (*Figure 3d*, $R^2$=0.9945 and 0.9973, respectively).

We then examined the performance of BLISA over cell lysates containing endogenously expressed GFP or mCherry. We used two drug-inducible gene expression systems (i.e., the tetracycline-regulated gene expression system and the cAMP response element-mediated gene expression system) to drive varying expression levels of GFP and mCherry in the cells (*Figure 3—figure supplement 1a and d*; see Materials and methods). We applied BLISA to simultaneously detect GFP/mCherry and α-tubulin (as the loading control). In parallel, we also conducted western blotting of GFP, mCherry, and α-tubulin on the same samples (*Figure 3—figure supplement 1b and e*; see Materials and methods). The detection results obtained from BLISA demonstrated high correlation with those from western blotting (*Figure 3—figure supplement 1c and f*). We obtained similar results when GFP and mCherry were co-expressed in the same cells (*Figure 3e–g*).

To further evaluate its multiplexity, we applied BLISA to simultaneously detect the phosphorylation of multiple target proteins. We implemented BLISA in a magnetic bead-based sandwich immunoassay format, wherein target proteins were immobilized by capture antibodies covalently attached to the magnetic beads and subsequently detected using a pool of DNA-barcoded detection antibodies (*Figure 4a*). We compared the sensitivity of magnetic bead-based BLISA with classic ELISA for detecting seven phosphorylated proteins (*Figure 4—figure supplement 1a and b*; see Materials and methods). The resulting standard curves demonstrated that BLISA exhibited either superior or comparable sensitivity to ELISA for all seven targets (*Figure 4—figure supplement 1a*). To assess the specificity of the magnetic bead-based BLISA, we prepared mixtures of the seven purified phosphorylated proteins at varying concentration gradients (*Figure 4—figure supplement 1c*). The BLISA-detected concentrations of all phosphorylated proteins closely aligned with their predetermined concentration gradients ($R^2$=0.9544, 0.9663, 0.9197, 0.9948, 0.9938, 0.9939, and 0.9198, respectively), with no significant cross-reactivity observed between different targets (*Figure 4—figure supplement 1c*).

Following the protocol validation, we applied the 7-plex magnetic bead-based BLISA to examine the effects of various treatments on the levels of seven phosphoproteins in U87 and U937 cells (non-treated, 10% fetal bovine serum [FBS] for 10 min, 100 nM anisomycin for 1 hr, and 20 mM $H_2O_2$ for 10 min) (*Figure 4b*). To facilitate a comparative analysis, cells were lysed and the resulting lysates were aliquoted for BLISA and ELISA (n=3 replicates for BLISA; n=2 replicates for each phosphoprotein and n=14 in total for ELISA). Results of the 7-plex BLISA were well correlated with ELISA (*Figure 4b*). Importantly, magnetic bead-based BLISA demonstrated superior multiplex capability by simultaneously detecting all phosphoproteins within individual samples. In contrast, ELISA necessitated separate aliquots for each phosphoprotein assay. This key distinction underscores BLISA's advantages in terms of efficiency, sample conservation, and cost-effectiveness.

A key feature of BLISA is its adaptability to a wide range of DNA-based technologies. We developed a high-throughput version of BLISA by incorporating next-generation sequencing (NGS) for simultaneous readout of DNA barcodes from multiple samples. We labeled individual antibody molecules with DNA oligos containing an antibody-specific 6 bp barcode sequence (Ab-bc) and a 15 bp unique molecular identifier (UMI) for quantification (*Islam et al., 2014*; *Figure 4—figure supplement 2a*). Following retrieval from antigen-antibody complexes via TCEP cleavage, the DNA barcodes were further tagged with a well-specific barcode (well-bc) through PCR amplification (*Figure 4—figure supplement 2a*). Subsequently, DNA barcodes from separate BLISA reactions were pooled together to prepare an indexed library for sequencing. This approach enables identification of each analyte in different samples through unique combinations of Ab-bcs and well-bcs, with quantification based on corresponding UMI counts. To validate the sequencing-based BLISA, we applied it to cell lysates spiked with varying concentrations of GFP and mCherry. The results demonstrated a high correlation between the sequencing-derived quantities of GFP and mCherry and their predetermined concentration gradients ($R^2$=0.9827 and 0.9843, respectively) (*Figure 4—figure supplement 2b and c*).

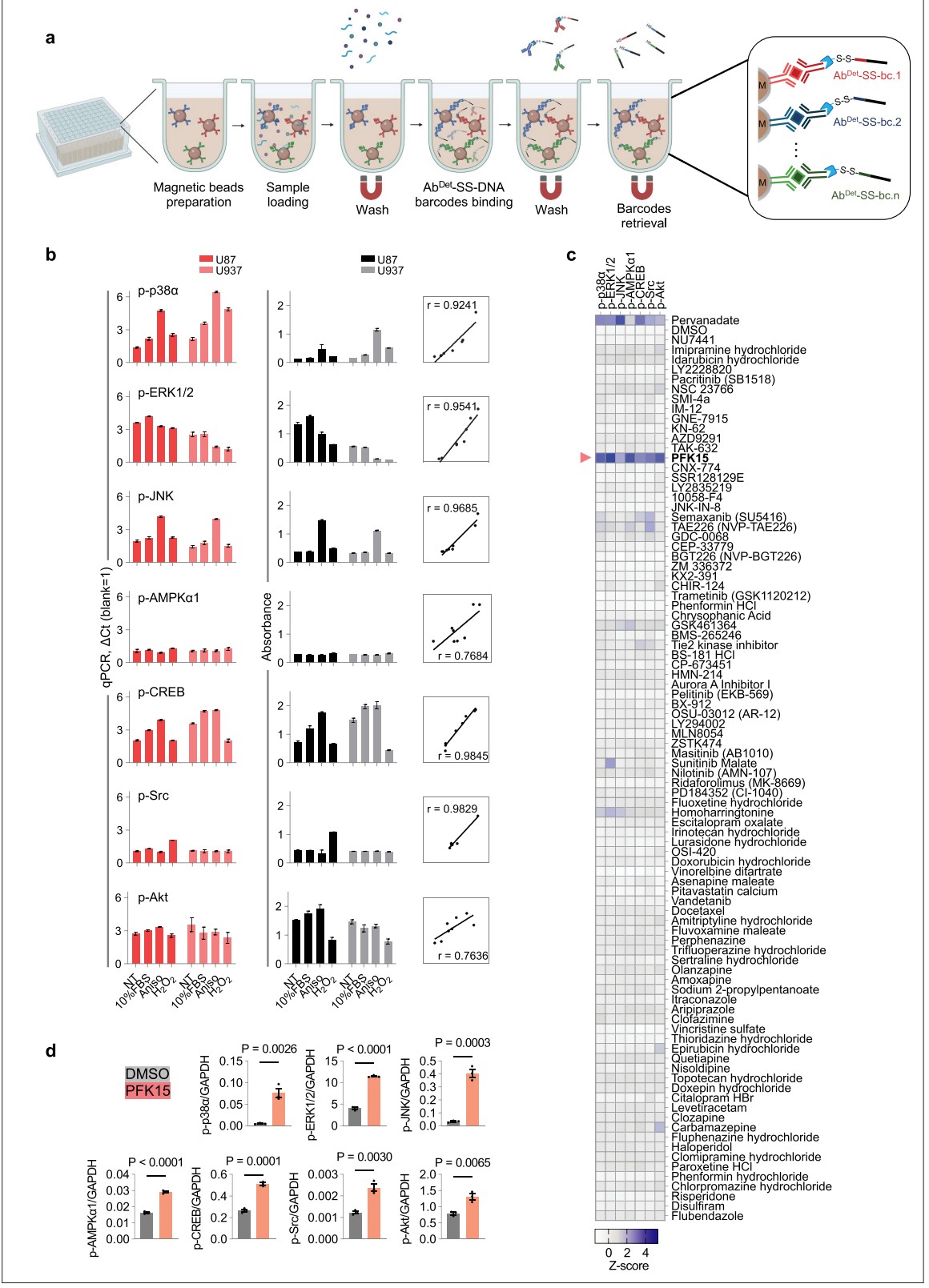

**Figure 4.** Magnetic bead-based barcode-linked immunosorbent assay (BLISA) for multiplexed detection. (**a**) Workflow of the magnetic bead-based BLISA. This panel was created using BioRender.com. (**b**) Comparison of 7-plex BLISA (left) and enzyme-linked immunosorbent assay (ELISA) (middle) for detecting the phosphorylation of seven endogenous proteins in U87 and U937 cells under various culture conditions. The right panel shows Pearson's correlation coefficients between BLISA and ELISA results for each target protein. n=3 replicates; mean ± s.e.m. (**c**) Heatmap showing average Z-scores

*Figure 4 continued on next page*

*Figure 4 continued*

of the phosphorylation levels of seven proteins in U87 cells following different drug treatments. n=2 biological replicates. (**d**) ELISA quantification of phosphorylation levels for seven proteins in U87 cells. n=3 biological replicates; two-sided t-tests. p-p38α, phospho-p38α (T180/Y182); p-ERK1/2, phospho-ERK1 (T202/Y204)/ERK2 (T185/Y187); p-JNK, phospho-JNK (Pan Specific); p-AMPKα1, phospho-AMPKα1 (T183); p-CREB, phospho-CREB (S133); p-Src, phospho-Src (Y419); p-Akt, phospho-Akt (S473); NT, non-treated; FBS, fetal bovine serum; Aniso, anisomycin.

The online version of this article includes the following figure supplement(s) for figure 4:

**Figure supplement 1.** Comparative analysis of standard curves for magnetic bead-based barcode-linked immunosorbent assay (BLISA) and enzyme-linked immunosorbent assay (ELISA).

**Figure supplement 2.** Validation of HTS-based barcode-linked immunosorbent assay (BLISA).

To demonstrate the potential of BLISA for high-throughput drug screening, we utilized an HTS-based 8-plex BLISA to the effects of 90 FDA-approved small-molecule compounds on U87 cells. The selected compounds were either kinase inhibitors exhibiting high blood-brain-barrier penetrance or drugs previously reported to inhibit GBM cell proliferation or migration (detailed compound information is listed in *Supplementary file 1*). Cells were treated with 10 µM of each compound or DMSO (vehicle control) for 24 hr. As a positive control, cells were treated with 1 mM pervanadate for 30 min. The magnetic bead-based BLISA was performed to simultaneously detect the phosphorylation of seven proteins and the level of GAPDH as a loading control. The screen was performed in duplicate. Among the 90 compounds, we identified PFK15 as a potent inducer for the phosphorylation of all seven target proteins (z-scores>2; *Figure 4c*). This finding was subsequently verified by individual ELISA specific to each target protein (*Figure 4d*). These results show that BLISA can utilize the NGS technology to dramatically increase the throughput.

## BLISA for high-throughput IgG detection in human serum

We asked whether BLISA could detect biomolecules in clinical samples with high throughput. The increasing threat posed by zoonotic diseases, as exemplified by the severe acute respiratory syndrome coronavirus 2 (SARS-CoV-2) pandemic, underscores the need for serologic surveillance methods to monitor population immunity (*Wajnberg et al., 2020*; *Khoury et al., 2021*). We applied the sequencing-based BLISA to human serum samples to detect IgG antibodies recognizing the SARS-CoV-2 spike receptor binding domain (RBD) (*Hoffmann et al., 2020*; *Lan et al., 2020*). We first compared the sensitivity of BLISA and the standard ELISA method for human IgG detection (*Figure 5—figure supplement 1a*). BLISA significantly outperforms ELISA, demonstrating approximately 49-fold higher sensitivity and an approximately 0.15 logs wider dynamic range (*Figure 5—figure supplement 1b*).

We then applied an indirect sandwich immunoassay to detect the anti-RBD IgGs in human serum samples: the anti-RBD IgGs were captured by the immobilized RBD proteins and were detected by the DNA-barcoded anti-human IgG antibody (*Figure 5a*). We first applied qPCR-based BLISA in serum samples from 24 donors who voluntarily disclosed their SARS-CoV-2 vaccination records. Of these, 23 had received at least one dose of an inactivated SARS-CoV-2 vaccine, while one donor had neither been vaccinated nor tested positive for SARS-CoV-2 (the negative control). We obtained anti-RBD IgG quantifications that were highly consistent with those acquired by ELISA (*Figure 5—figure supplement 1c*). The anti-RBD IgG level of the negative control sample was indeed lower than those from vaccinated donors (*Figure 5—figure supplement 1c*). Considerable variations in the anti-RBD IgG levels were observed among samples from vaccinated donors (*Figure 5—figure supplement 1c*), indicating differential vaccination status and/or immune responses. Subsequently, we conducted HTS-based BLISA on a cohort of 517 human serum samples (including the 24 serum samples), the majority of which (n=493) were collected from healthy individuals undergoing routine physical examinations during the initial phase of the SARS-CoV-2 vaccination campaign in Beijing, China. Each serum sample was assayed in duplicates (1034 reactions in separated wells; twelve 96-well plates). As a technical control, 12 samples from vaccinated donors were randomly selected and distributed across 12 different BLISA plates. For each plate, we also included the negative control sample as well as a blank sample (without serum input). To mitigate sample-to-sample variations potentially introduced during sequencing library preparation, we spiked additional normalization DNA oligos into samples during the barcode retrieval step (*Figure 5a*; *Figure 5—figure supplement 2a and b*; see Materials and methods). The variation between duplicates was substantially reduced after normalization, with 98.4% of samples exhibiting a coefficient of variation (CV) below 0.2 (*Figure 5—figure supplement 3a*).

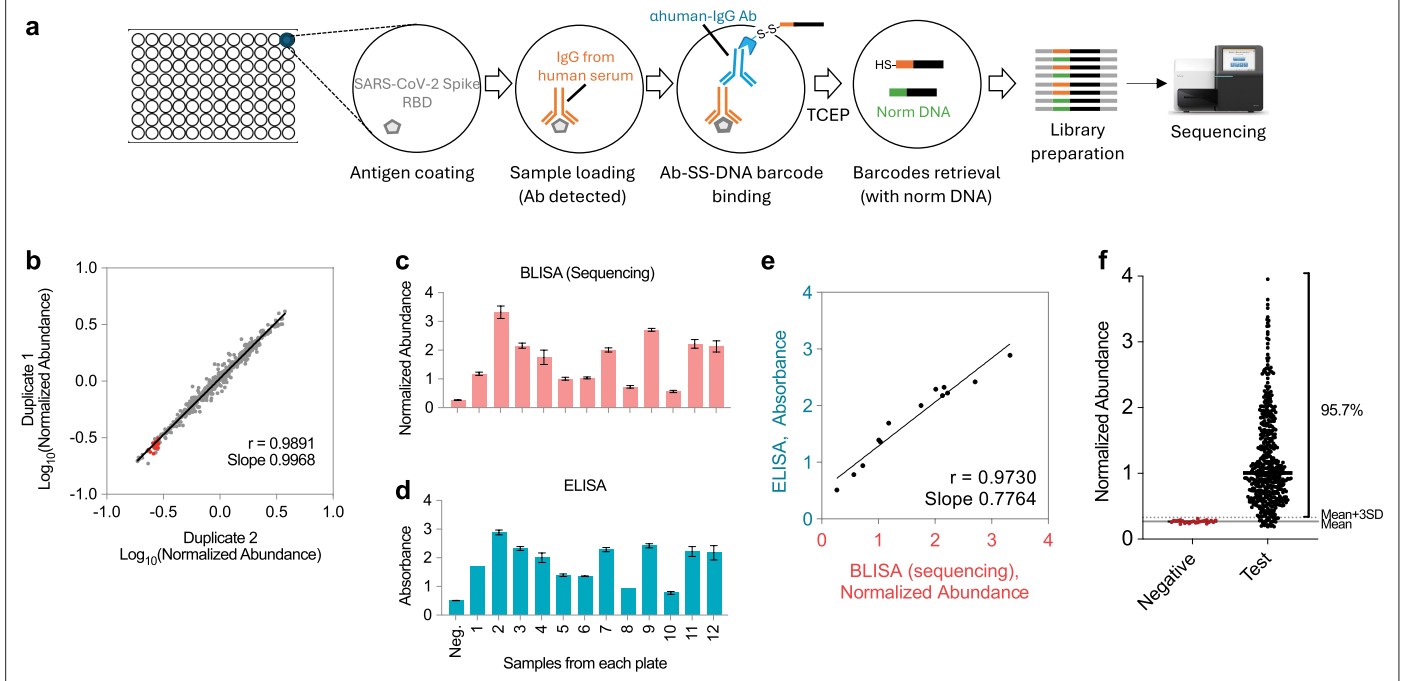

**Figure 5.** HTS-based barcode-linked immunosorbent assay (BLISA) for high-throughput detection of IgGs in human serum samples. (**a**) Workflow of sandwich immunoassay-based IgG detection using BLISA. (**b**) Pearson's correlation between results from two technical duplicates. Red dots represent negative samples. (**c,d**) Quantification of anti-receptor binding domain (RBD) IgG levels in 12 samples from vaccinated donors using HTS-based BLISA (**c**) and enzyme-linked immunosorbent assay (ELISA) (**d**). n = 2 replicates; mean ± s.e.m. (**e**) Pearson's correlation between BLISA and ELISA results shown in (**c**) and (**d**). (**f**) Distribution of normalized IgG abundance in negative and test samples (n=493 samples). Red dots represent negative samples. The solid gray line indicates the mean normalized abundance of negative samples. The dotted gray line represents the mean plus three standard deviations (mean + 3 SD).

The online version of this article includes the following figure supplement(s) for figure 5:

**Figure supplement 1.** Comparative analysis of the sensitivity of barcode-linked immunosorbent assay (BLISA) and enzyme-linked immunosorbent assay (ELISA) for human IgG detection.

**Figure supplement 2.** Sequencing library preparation for barcode-linked immunosorbent assay (BLISA).

**Figure supplement 3.** Quantification of anti-receptor binding domain (RBD) IgG in human serum samples using HTS-based barcode-linked immunosorbent assay (BLISA).

No significant plate-to-plate variation in the detection results was observed (*Figure 5—figure supplement 3b and c*). The quantifications by HTS-based BLISA were highly consistent between technical duplication (*Figure 5b*). The anti-RBD IgG levels of the 12 samples from vaccinated donors quantified by HTS-based BLISA were again highly consistent with ELISA measurements (*Figure 5c–e*). Notably, the anti-RBD IgG levels of the majority of assayed samples (95.7%, measured using a threshold at three standard deviations above the mean of the negative samples) were higher than those of the negative samples (*Figure 5f*). This result aligns with the high vaccination rate during the time period in Beijing, China, when the samples were collected.

## High-throughput 2-plex BLISA for human HBV diagnosis

We performed 2-plex BLISA for the simultaneous detection of hepatitis B surface antigen (HBsAg) and hepatitis B e antigen (HBeAg) in serum samples. These two serological markers are crucial for the diagnosis and monitoring of hepatitis B virus (HBV) infection. The significant difference in the physiological concentrations of these antigens in blood (HBsAg being substantially more abundant than HBeAg; *Kessler and Jimenez, 2019*) presents a considerable challenge for concurrent detection. We again utilized a sandwich immunoassay format: both antigens (HBsAg and HBeAg) were captured by their respective immobilized capture antibodies (Ab$^{Cap}$) and subsequently detected using DNA-barcoded detection antibodies (Ab$^{Det}$-SS-DNA barcode) (*Figure 6a*). For the initial test, we obtained a cohort of 529 human serum samples (designated as cohort 1). These samples were analyzed in

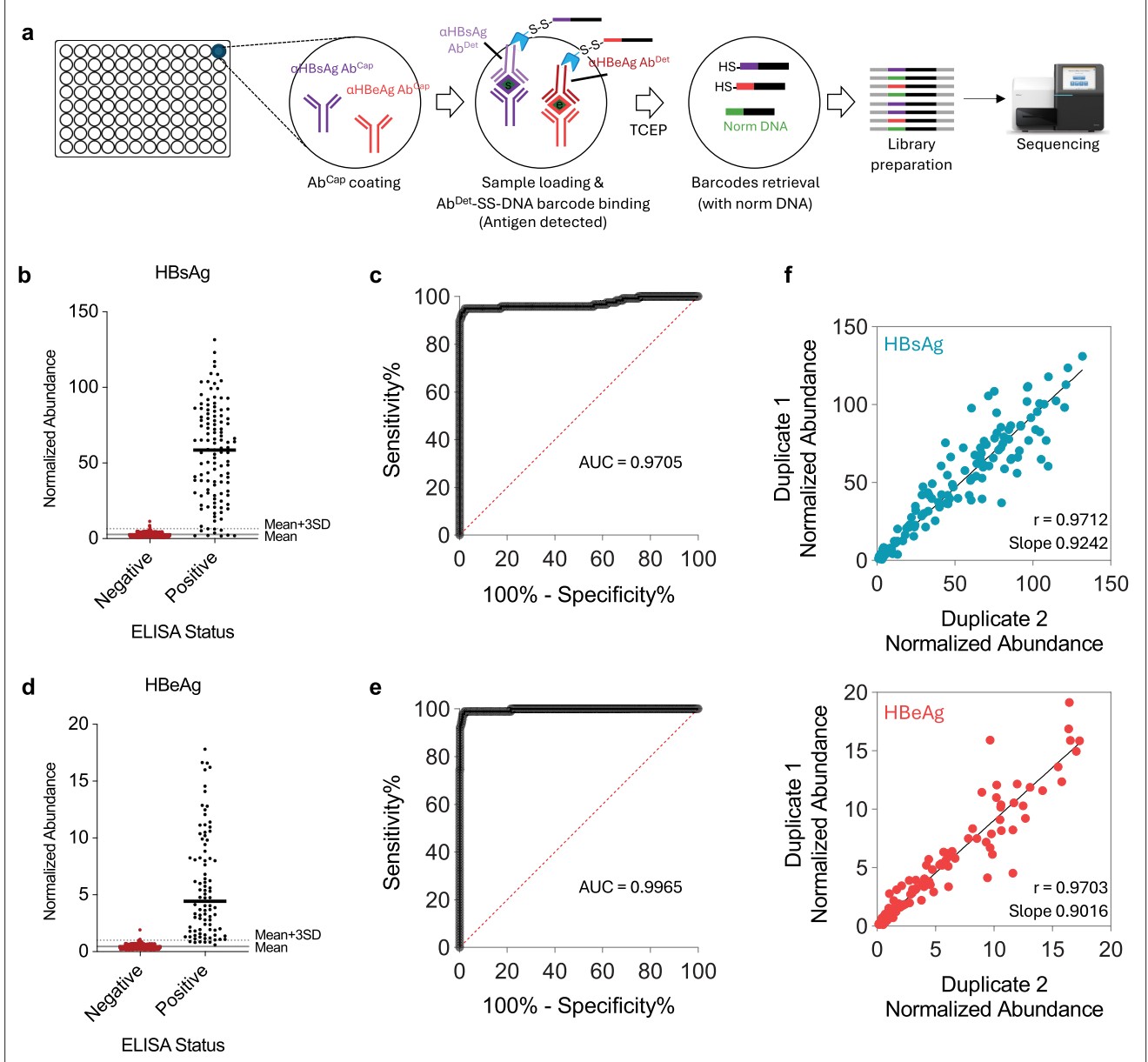

**Figure 6.** HTS-based barcode-linked immunosorbent assay (BLISA) for high-throughput simultaneous detection of two hepatitis B virus (HBV) antigens in human serum samples. (**a**) Workflow of sandwich immunoassay-based HBV antigen detection using BLISA. (**b, c**) Sensitivity and specificity of 2-plex BLISA for detecting hepatitis B surface antigen (HBsAg). n=462 enzyme-linked immunosorbent assay (ELISA)-negative samples from cohort 2; n=119 ELISA-positive samples from cohort 2. (**d, e**) Sensitivity and specificity of 2-plex BLISA on detecting hepatitis B e antigen (HBeAg). n=505 ELISA-negative samples from cohort 2; n=90 ELISA-positive samples from cohort 2. Solid gray lines indicate the mean normalized abundance of ELISA-confirmed negative samples. Dotted gray lines represent the mean plus three standard deviations (mean + 3 SD). (**f**) Pearson's correlations between technical duplicates for HBsAg (upper) and HBeAg (bottom) measurements in serum samples from cohort 2 (n=600).

The online version of this article includes the following figure supplement(s) for figure 6:

**Figure supplement 1.** HTS-based barcode-linked immunosorbent assay (BLISA) for high-throughput detection of two hepatitis B virus (HBV) antigens in serum samples from cohort 1.

**Figure supplement 2.** Performance evaluation of 2-plex barcode-linked immunosorbent assay (BLISA) for detecting hepatitis B virus (HBV) antigens using enzyme-linked immunosorbent assay (ELISA)-independent thresholds.

parallel (in duplicate; 1058 reactions in separated wells on twelve 96-well plates) (*Figure 6—figure supplement 1a*). We employed differential concentrations of detection antibodies: anti-HBsAg at 0.33 nM and anti-HBeAg at 0.033 nM, as recommended by the manufacturer (detailed antibody information in *Supplementary file 2*). This 10-fold difference in antibody concentration resulted in significant variations in read number distribution between HBsAg and HBeAg in the sequencing data (*Figure 6—figure supplement 1b*). Importantly, despite these differences, the simultaneous detection of these two antigens within the same BLISA remained unaffected. For both the HBsAg and HBeAg, the detections were highly consistent between the duplicate (r=0.9973, p<0.0001; r=0.9770, p<0.0001) (*Figure 6—figure supplement 1c*). The levels of both HBsAg and HBeAg detected by BLISA were significantly higher in the positive control samples than in the negative control samples (*Figure 6—figure supplement 1d*).

To validate our results, we performed ELISA on a subset of 115 samples and identified 28 HBsAg[+] samples and 7 HBeAg[+] samples (*Figure 6—figure supplement 1e and g*). We analyzed the sensitivity and specificity of BLISA on this ELISA-confirmed subset. The threshold for positive sample identification in the BLISA dataset was set at three standard deviations above the mean of the ELISA-confirmed negative samples (*Figure 6—figure supplement 1e and g*). For HBsAg, BLISA achieved a sensitivity of 100% (95% CI, 87.94–100.0%) and a specificity of 96.55% (95% CI, 90.35–99.06%) in discriminating ELISA-confirmed negative and positive samples (*Figure 6—figure supplement 1e and f*). For HBeAg, BLISA achieved a sensitivity of 100% (95% CI, 64.57–100.0%) and a specificity of 96.30% (95% CI, 90.86–98.55%) in discriminating ELISA-confirmed negative and positive samples (*Figure 6—figure supplement 1g and h*).

We next collected a new cohort of serum samples from 600 donors (designated as cohort 2). To ensure enough positive samples for statistical analysis, this new cohort included 100 samples from donors that have been clinically diagnosed as HBV positive. To establish a reference standard for analyzing the specificity and sensitivity of BLISA, we performed ELISA on all samples and identified 119 HBsAg[+] samples and 90 HBeAg[+] samples (*Figure 6b and d*). Subsequently, we performed HTS-based BLISA on this new cohort (in duplicate; 1200 reactions in separated wells on fourteen 96-well plates). The detections of both the HBsAg and HBeAg were again highly consistent between the duplicate (r=0.9712, p<0.0001; r=0.9703, p<0.0001) (*Figure 6f*). For HBsAg, BLISA achieved a sensitivity of 92.4% (95% CI, 86.25–95.97%) and a specificity of 99.1% (95% CI, 97.80–99.66%) in discriminating ELISA-confirmed negative and positive samples (*Figure 6b and c*). For HBeAg, BLISA achieved a sensitivity of 93.3% (95% CI, 86.21–96.91%) and a specificity of 99.6% (95% CI, 98.57–99.93%) in discriminating ELISA-confirmed negative and positive samples (*Figure 6d and e*).

We conducted additional analyses wherein the thresholds for positive sample identification in the BLISA datasets were established independently of ELISA data. Considering the low prevalence of HBV infection in the general Chinese population, recently estimated at 3% (95% CI: 2–4%) (*Bai et al., 2024*), we assumed that the majority of samples in our two cohorts would be negative for HBsAg and HBeAg. Thus, we set the threshold at one standard deviation above the mean of the entire cohorts for positive sample identification in the BLISA dataset (*Figure 6—figure supplement 2*). We note that the ELISA-independent approach, while effective in low-prevalence populations, may be unsuitable for high-prevalence populations. We calculated the threshold for cohort 1 (529 samples) and applied it to the 115 ELISA-confirmed samples in cohort 1 (*Figure 6—figure supplement 2a and b*). For HBsAg, BLISA achieved a sensitivity of 92.86% (95% CI, 77.35–98.73%) and a specificity of 100% (95% CI, 95.77–100%) in discriminating ELISA-confirmed negative and positive samples (*Figure 6—figure supplement 2a*). For HBeAg, BLISA achieved a sensitivity of 100% (95% CI, 64.57–100%) and a specificity of 94.44% (95% CI, 88.41–97.43%) in discriminating ELISA-confirmed negative and positive samples (*Figure 6—figure supplement 2b*). Similarly, we calculated the threshold for cohort 2 (based on 500 samples in cohort 2, excluded 100 positive control samples; *Figure 6—figure supplement 2c and d*). For HBsAg, BLISA achieved a sensitivity of 90.76% (95% CI, 84.20–94.76%) and a specificity of 99.78% (95% CI, 98.78–99.99%) in discriminating ELISA-confirmed negative and positive samples (*Figure 6—figure supplement 2c*). For HBeAg, BLISA achieved a sensitivity of 91.11% (95% CI, 83.43–95.43%) and a specificity of 99.80% (95% CI, 98.89–99.99%) in discriminating ELISA-confirmed negative and positive samples (*Figure 6—figure supplement 2d*). These results thus demonstrate that BLISA enables multiplexed and high-throughput detection of disease-related biomolecules in clinical samples with high specificity and sensitivity.

## Discussion

DNA-barcoded antibodies have been increasingly utilized to unravel the molecular characteristics of the crowded biological environment (*Lewis et al., 2021*; *van Buggenum et al., 2018*; *Stoeckius et al., 2017*; *Hwang et al., 2021*; *Vistain et al., 2022*; *Saka et al., 2019*; *Goltsev et al., 2018*; *Lundberg et al., 2011*). However, despite the extensive repertoire of commercially available antibodies, a universally applicable strategy for site-specific antibody-DNA conjugation remains elusive. Previous approaches often require custom conjugation protocols for individual antibodies and specific applications, resulting in labor-intensive and costly processes (*Hickey et al., 2022*). Furthermore, traditional non-site-specific approaches, which rely on random conjugation to specific amino acid residues (e.g., lysine or cysteine), often compromise antibody affinity and specificity due to potential modification of residues within or adjacent to antigen-binding sites. To address these limitations, site-directed antibody-DNA conjugation approaches have been developed by utilizing the non-canonical amino acids (*Kazane et al., 2012*) or affinity reagents (e.g., nitrilotriacetic acid [*Rosen et al., 2014*], FCIII peptide [*Nielsen et al., 2019*], protein G [*Cremers et al., 2019*], or the Z domain of protein A [*Stiller et al., 2019*]). Nonetheless, these techniques typically yield low product quantities and necessitate multiple components or genetic reengineering of the antibody.

The MaMBA strategy, developed in this study, offers a highly efficient and site-specific DNA barcoding approach that is broadly applicable to IgG antibodies. Classic antibody conjugation methods necessitate stringent reaction conditions to achieve optimal yields, typically requiring a minimum of 50–100 µg IgG antibodies at 1 mg/mL per reaction. For commercially available antibodies, the buffer recipe and the stock concentration vary significantly across manufacturers and may not be compatible with the conjugation reactions. Consequently, buffer exchange and antibody concentration adjustments are often necessary prior to the reaction. Additionally, both antibodies and DNA oligos require activation for subsequent conjugation, with the activated intermediates being unstable and necessitating fresh preparation before each reaction. These requirements collectively increase operational complexity and costs (*Figure 1—figure supplement 6*). By utilizing anti-IgG Fc nanobodies (*Pleiner et al., 2018*) as adaptors, MaMBA achieves a remarkable degree of modularity. The DNA oligos are site-specifically conjugated to the C-terminal of the anti-IgG Fc nanobodies through a highly efficient enzymatic reaction by OaAEP1(C247A). This enzyme exhibits high catalytic efficiency and strict sequence specificity, recognizing a minimal NGL tripeptide motif at nanobodies C-termini. The reaction proceeds through a thioester intermediate formed after cleavage at the asparagine residue, followed by nucleophilic attack (*Yang et al., 2017*; *Rehm et al., 2019*; *Harris et al., 2015*). Using a GV-dipeptide-based acceptor nucleophiles (i.e., the GVG-K($N_3$)-RG polypeptides), we achieved stable azide-modified nanobodies production. The well-defined sequence specificity of OaAEP1(C247A) enables efficient and site-specific conjugation, minimizing potential impacts on nanobody functionality while enabling large-scale production of DNA-nanobody barcoding modules with diverse DNA sequences. The monovalent binding nature of the anti-Fc nanobodies also simplifies the barcoding procedure to a single mixing step, thus offering significantly advantageous applicability. For proof-of-principle demonstrations, we selected two well-characterized IgG secondary nanobodies: TP897 (rabbit IgG-specific) and TP1107 (mouse IgG1-specific). These nanobodies demonstrate exceptional binding affinities with dissociation constants ($K_d$) in the nanomolar to picomolar range (*Bae et al., 2020*). Their reliability and accessibility, with sequences and plasmids publicly available and proteins commercially available, made them ideal candidates for validating MaMBA's capability and accuracy. The broad applicability of these nanobodies to commonly used mouse and rabbit-derived commercial IgGs further supports their utility as molecular adaptors. Additionally, the compatibility of MaMBA extends beyond TP897 and TP1107. A broader series of high-affinity IgG secondary nanobodies, previously reported by *Pleiner et al., 2018*, offers diverse characteristics, including varied species specificity, epitope recognition, and subclass specificity. This expanded toolkit should enable wide-ranging applications of MaMBA across different IgG types, enhancing its versatility as an antibody-DNA barcoding platform. Thus, MaMBA shall promote the widespread adoption of DNA-barcoded antibody-based technologies across a multitude of biomedical and diagnostic applications.

To demonstrate the utility of MaMBA, we employed this method to develop novel techniques for multi-channel in situ protein imaging (*mis*HCR$^n$) and high-throughput biomolecule detections (BLISA). The MaMBA-based *mis*HCR$^n$ method enables in situ imaging of biomolecules with enhanced multiplicity and sensitivity compared to traditional immunohistochemistry, which is typically limited in the number

of detectable biomolecules due to spectral and host species constraints. The *mis*HCR$^n$ method leverages DNA hairpin chain reaction (HCR), where metastable fluorophore-labeled DNA hairpins (amplifiers) undergo chain hybridization triggered by specific initiator sequences, creating long polymer chains that amplify detection signals (*Choi et al., 2014*). We initially developed the isHCR method to extend the multiplexing capability of immunohistochemistry through the use of HCR initiators with distinct DNA sequences (*Lin et al., 2018*). However, the potential of isHCR remains constrained by the limited availability of orthogonal HCR initiator/amplifier pairs (i.e., five pairs from the original HCR study and eight additional newly reported pairs; *Wang et al., 2024*). MaMBA addresses these limitations by directly attaching HCR initiators to primary antibodies via a cleavable chemical linker, thereby simplifying the *mis*HCR$^n$ staining procedure and eliminating host species overlaps. Moreover, MaMBA enables the removal of HCR initiators after image acquisition using a mild reducing agent (e.g., TCEP; *Pham et al., 2021*), allowing for iterative uses of HCR initiator/amplifier pairs. We achieved 12-plex *mis*HCR$^n$ immunodetection on mouse brain sections using 12 rabbit IgGs and a standard confocal imaging setup. The multiplexity and the imaging throughput of *mis*HCR$^n$ can be further enhanced by incorporating advanced chemical probes and imaging techniques, such as Raman dyes (*Wei et al., 2017*; *Shi et al., 2022*) and spectral unmixing (*Seo et al., 2022*). As a versatile antibody barcoding technique, the utility of MaMBA in immunohistochemistry extends beyond our demonstrations with *mis*HCR$^n$ here. MaMBA could be readily employed in various existing DNA-based multiplexed immunofluorescence methods, including DNA-Exchange (DEI) (*Wang et al., 2017*; *Jungmann et al., 2014*), CODEX (*Goltsev et al., 2018*; *Black et al., 2021*), and immuno-SABER (*Saka et al., 2019*).

Leveraging the flexibility of the ELISA design, we utilized MaMBA to develop BLISA for multiplexed and quantitative biomolecule detections. Traditional ELISA typically relies on arrayed enzymatic reactions to report the quantities of target molecules in different samples. In contrast, MaMBA-based BLISA replaces the enzyme reporters with retrievable single-stranded DNA barcodes, enabling quantifications via qPCR or NGS technologies in a highly multiplexed and high-throughput way. Moreover, BLISA is compatible with various sample types (e.g., serum and cell lysate) and assay formats (e.g., plate- or beads-based assay). We successfully used BLISA to simultaneously measure the levels of anti-SARS-CoV-2-RBD antibody or HBV antigens in over 1000 human serum samples. Moreover, we demonstrated an HTS-based 8-plex BLISA for a small-scale drug screen. Representing a high-throughput platform with dramatically enhanced multiplex capacity, BLISA effectively lowers the costs, sample-volume requirement, and turnaround time. Lab automation for sample handling and sequencing library preparation could further facilitate the adoption of BLISA in large-scale applications in both clinical diagnostics and translation research (e.g., pandemic surveillance and biomarker discoveries). Additionally, other existing methods, including immuno-PCR (*Sano et al., 1992*; *Niemeyer et al., 2007*), ID-seq (*van Buggenum et al., 2018*; *van Buggenum et al., 2016*), the proximity extension assay (*Fredriksson et al., 2002*; *Wik et al., 2021*), and single-cell sequencing-based protein profiling (*Stoeckius et al., 2017*; *Hwang et al., 2021*; *Chung et al., 2021*; *Mimitou et al., 2019*), would also benefit from our MaMBA approach.

## Materials and methods

**Key resources table**

| Reagent type (species) or resource | Designation | Source or reference | Identifiers | Additional information |
|---|---|---|---|---|
| Strain, strain background (*Escherichia coli*) | SHuffe | Weidibio | Cat#EC2031 | |
| Strain, strain background (*Escherichia coli*) | BL21 (DE3) | TransGen Biotech | Cat#CD601 | |
| Strain, strain background (*Mus musculus*) | C57BL/6N | Beijing Vital River | RRID:MGI:2159965 | |
| Cell line (*Homo sapiens*) | HEK293T | ATCC | CRL-3216; RRID:CVCL_0063 | |

*Continued on next page*

*Continued*

| Reagent type (species) or resource | Designation | Source or reference | Identifiers | Additional information |
|---|---|---|---|---|
| Cell line (*Homo sapiens*) | HeLa | ATCC | CCL-2; RRID:CVCL_0030 | |
| Cell line (*Homo sapiens*) | U-87 MG | ATCC | HTB-14; RRID:CVCL_0022 | |
| Cell line (*Homo sapiens*) | U-937 | ATCC | CRL-1593.2; RRID:CVCL_0007 | |
| Chemical compound, drug | DBCO-PEG$_3$-SS-NHS | Conju-Probe | Cat#CP-2089; CAS: 2163772-16-3 | |
| Chemical compound, drug | Anhydrotetracycline hydrochloride (ATc) | J&K | Cat#541642; CAS: 13803-65-1 | |
| Chemical compound, drug | Forskolin (FSK) | Sigma-Aldrich | Cat#F3917; CAS: 66575-29-9 | |
| Chemical compound, drug | 3-Isobutyl-1-methylxanthine (IBMX) | Sigma-Aldrich | Cat#I5879; CAS: 28822-58-4 | |
| Chemical compound, drug | Anisomycin | MedChem Express | Cat#CS-4981; CAS: 22862-76-6 | |
| Chemical compound, drug | Compounds in HTS-based BLISA | Other | N/A | See *Supplementary file 1* |
| Transfected construct | P$_{Tight}$-EGFP-PGK-rtTA | This paper | N/A | See *Supplementary file 4* |
| Transfected construct | 3×CRE-mCherry-PEST | This paper | N/A | See *Supplementary file 4* |
| Biological sample (*Homo sapiens*) | Human healthy (male) and psoriasis (female) skins | Other | N/A | From Ting Chen's Lab; see Materials and methods |
| Biological sample (*Homo sapiens*) | Human serum | Beijing, China | N/A | See Materials and methods |
| Antibody | anti-PDGFRα (Rabbit monoclonal) | Abcam | Cat#ab203491; RRID:AB_2892065 | 1:500 |
| Antibody | anti-KRT14 (Rabbit monoclonal) | https://doi.org/10.7554/eLife.52712 | N/A | Gift from Ting Chen's Lab; 1:1000 |
| Antibody | anti-DCT (Rabbit monoclonal) | https://doi.org/10.7554/eLife.52712 | N/A | Gift from Ting Chen's Lab; 1:500 |
| Antibody | anti-CD31 (Mouse monoclonal) | Abcam | Cat#ab9498; RRID:AB_307284 | 1:500 |
| Antibody | anti-αSMA (Rabbit monoclonal) | Abcam | Cat#ab124964; RRID:AB_11129103 | 1:500 |
| Antibody | anti-CD45 (Mouse monoclonal) | BD Biosciences | Cat#555481; RRID:AB_395873 | 1:300 |
| Antibody | anti-MAP2 (Rabbit polyclonal) | Thermo Fisher Scientific | Cat#PA5-85755; RRID:AB_2792892 | 1:500 |
| Antibody | anti-NeuN (Rabbit monoclonal) | Abcam | Cat#ab177487; RRID:AB_2532109 | 1:500 |
| Antibody | anti-TPH2 (Rabbit monoclonal) | Abcam | Cat#ab184505; RRID:AB_2892828 | 1:200 |
| Antibody | anti-TH (Rabbit monoclonal) | Millipore | Cat#AB152; RRID:AB_390204 | 1:500 |
| Antibody | anti-nNOS (Rabbit monoclonal) | Sigma-Aldrich | Cat#N7280; RRID:AB_260796 | 1:500 |
| Antibody | anti-GFAP (Rabbit monoclonal) | Abcam | Cat#ab7260; RRID:AB_305808 | 1:750 |
| Antibody | anti-Iba1 (Rabbit monoclonal) | FUJIFILM Wako | Cat#019–19741; RRID:AB_839504 | 1:500 |

*Continued on next page*

*Continued*

| Reagent type (species) or resource | Designation | Source or reference | Identifiers | Additional information |
|---|---|---|---|---|
| Antibody | anti-DDC (Rabbit polyclonal) | Abcam | Cat#ab3905; RRID:AB_304145 | 1:50 |
| Antibody | anti-NF-H (Rabbit monoclonal) | Thermo Fisher Scientific | Cat#711025; RRID:AB_2609477 | 1:100 |
| Antibody | anti-TMEM119 (Rabbit monoclonal) | Abcam | Cat#ab209064; RRID:AB_2800343 | 1:500 |
| Antibody | anti-GABA (Rabbit polyclonal) | Sigma-Aldrich | Cat#A2025; RRID:AB_3698410 | 1:750 |
| Antibody | anti-Orexin A (Rabbit polyclonal) | Phoenix Pharmaceuticals | Cat#H-003–30; RRID:AB_2315019 | 1:500 |
| Antibody | anti-5-HT (Rabbit polyclonal) | Immunostar | Cat#20080; RRID:AB_572263 | 1:500 |
| Antibody | anti-$\alpha$-tubulin (Mouse monoclonal) | Sigma-Aldrich | Cat#T5168; RRID:AB_477579 | 1:40,000 |
| Antibody | anti-GFP (Rabbit polyclonal) | Thermo Fisher Scientific | Cat#A-11122; RRID:AB_221569 | 1:1000 |
| Antibody | anti-mCherry (Rabbit polyclonal) | Thermo Fisher Scientific | Cat#PA5-34974; RRID:AB_2552323 | 1:2000 for purified mCherry, 1:10,000 for the expressed mCherry |
| Antibody | anti-Human IgG (Rabbit monoclonal) | Abcam | Cat#ab181236; RRID:AB_3698401 | 0.33 nM |
| Antibody | anti-HBsAg (capture) (Goat monoclonal) | Beijing Wantai Biological | Cat#YTX2101 | 5 µg/mL |
| Antibody | anti-HBsAg (detection) (Mouse monoclonal) | Beijing Wantai Biological | Cat#HBs-2C1 | 0.33 nM |
| Antibody | anti-HBeAg (capture) (Mouse monoclonal) | Beijing Wantai Biological | Cat#13B12-1 | 5 µg/mL |
| Antibody | anti-HBeAg (detection) (Mouse monoclonal) | Beijing Wantai Biological | Cat#9A4-1 | 0.033 nM |
| Antibody | anti-phospho-p38$\alpha$ (T180/Y182) (capture) (Mouse monoclonal) | R&D Systems | Cat#DYC869B; RRID:AB_3698412 | The vendor recommended conc. |
| Antibody | anti-phospho-p38$\alpha$ (T180/Y182) (detection) (Rabbit monoclonal) | R&D Systems | Cat#DYC869B; RRID:AB_3698412 | 1/10 of the vendor recommended conc. |
| Antibody | anti-phospho-ERK1 (T202/Y204)/ ERK2 (T185/Y187) (capture) (Mouse monoclonal) | R&D Systems | Cat#DYC1018B; RRID:AB_3698414 | The vendor recommended conc. |
| Antibody | anti-phospho-ERK1 (T202/Y204)/ ERK2 (T185/Y187) (detection) (Rabbit monoclonal) | R&D Systems | Cat#DYC1018B; RRID:AB_3698414 | 1/10 of the vendor recommended conc. |
| Antibody | anti-phospho-JNK Pan Specific (capture) (Mouse monoclonal) | R&D Systems | Cat#DYC1387B; RRID:AB_3698415 | The vendor recommended conc. |
| Antibody | anti-phospho-JNK Pan Specific (detection) (Rabbit monoclonal) | R&D Systems | Cat#DYC1387B; RRID:AB_3698415 | 1/10 of the vendor recommended conc. |
| Antibody | anti-phospho-AMPK$\alpha$1 (T183) (capture) (Goat monoclonal) | R&D Systems | Cat#DYC3528; RRID:AB_3698416 | The vendor recommended conc. |
| Antibody | anti-phospho-AMPK$\alpha$1 (T183) (detection) (Rabbit monoclonal) | R&D Systems | Cat#DYC3528; RRID:AB_3698416 | 1/10 of the vendor recommended conc. |
| Antibody | anti-phospho-CREB (S133) (capture) (Goat monoclonal) | R&D Systems | Cat#DYC2510; RRID:AB_3698417 | The vendor recommended conc. |
| Antibody | anti-phospho-CREB (S133) (detection) (Rabbit monoclonal) | R&D Systems | Cat#DYC2510; RRID:AB_3698417 | 1/10 of the vendor recommended conc. |

*Continued on next page*

*Continued*

| Reagent type (species) or resource | Designation | Source or reference | Identifiers | Additional information |
|---|---|---|---|---|
| Antibody | anti-phospho-Src (Y419) (capture) (Goat monoclonal) | R&D Systems | Cat#DYC2685; RRID:AB_3698418 | The vendor recommended conc. |
| Antibody | anti-phospho-Src (Y419) (detection) (Rabbit monoclonal) | R&D Systems | Cat#DYC2685; RRID:AB_3698418 | 1/10 of the vendor recommended conc. |
| Antibody | anti-phospho-Akt (S473) (capture) (Rabbit monoclonal) | Abcam | Cat#ab285034; RRID:AB_3698403 | 6.0 µg/mL |
| Antibody | anti-phospho-Akt (S473) (detection) (Rabbit monoclonal) | Abcam | Cat#ab285140; RRID:AB_3698406 | 0.01 µg/mL |
| Recombinant DNA reagent | pET28a-His$_6$-Ubiquitin-OaAEP1 (C247A) | https://doi.org/10.1038/ncomms10199 | N/A | See *Supplementary file 4* |
| Recombinant DNA reagent | pET21a-TP897-NGL-His$_6$ | This paper | N/A | See *Supplementary file 4* |
| Recombinant DNA reagent | pET21a-TP1107-NGL-His$_6$ | This paper | N/A | See *Supplementary file 4* |
| Recombinant DNA reagent | pET28a-His$_6$-EGFP | This paper | N/A | See *Supplementary file 4* |
| Recombinant DNA reagent | pET28a-His$_6$-mCherry | This paper | N/A | See *Supplementary file 4* |
| Sequence-based reagent | HCR Initiators | https://doi.org/10.1021/nn405717p | HCR probes | See *Supplementary file 3* |
| Sequence-based reagent | HCR Amplifiers | https://doi.org/10.1021/nn405717p | HCR probes | See *Supplementary file 3* |
| Sequence-based reagent | DNA barcode oligos for qPCR | This paper | DNA barcodes | See *Supplementary file 3* |
| Sequence-based reagent | DNA barcode oligos for sequencing | This paper | DNA barcodes | See *Supplementary file 3* |
| Sequence-based reagent | Primers for qPCR-based BLISA | This paper | qPCR primers | See *Supplementary file 5* |
| Peptide, recombinant protein | GVG-K(N$_3$)-RG | Scilight-Peptide | N/A | |
| Peptide, recombinant protein | OaAEP1 (C247A) | This paper | N/A | See Materials and methods |
| Peptide, recombinant protein | TP897-NGL-His$_6$ | This paper | N/A | See Materials and methods |
| Peptide, recombinant protein | TP1107-NGL-His$_6$ | This paper | N/A | See Materials and methods |
| Peptide, recombinant protein | GFP | This paper | N/A | See Materials and methods |
| Peptide, recombinant protein | mCherry | This paper | N/A | See Materials and methods |
| Commercial assay or kit | BCA protein assay | Thermo Fisher Scientific | Cat#23225 | |
| Software, algorithm | MATLAB | MathWorks | vR2018a; RRID:SCR_001622 | |
| Software, algorithm | GraphPad Prism | GraphPad | v9.02; RRID:SCR_002798 | |
| Software, algorithm | Zen | Zeiss | v2.3; RRID:SCR_013672 | |
| Software, algorithm | LAS X | Leica | v5.1; RRID:SCR_013673 | |
| Software, algorithm | ImageJ | NIH | v 2.1.0; RRID:SCR_003070 | |

*Continued on next page*

*Continued*

| Reagent type (species) or resource | Designation | Source or reference | Identifiers | Additional information |
|---|---|---|---|---|
| Software, algorithm | Python | The Python Software Foundation | 3.8.10 | https://www.python.org/downloads/ |
| Software, algorithm | UMI-tools | *Smith et al., 2017* | v1.1.2; RRID:SCR_017048 | https://github.com/CGATOxford/UMI-tools |
| Software, algorithm | Snakemake | *Mölder et al., 2021* | v6.8.0; RRID:SCR_003475 | https://github.com/snakemake/snakemake |
| Software, algorithm | Biopython | *Cock et al., 2009* | v1.79; RRID:SCR_007173 | https://github.com/biopython/biopython |
| Software, algorithm | R | R Core Team | v4.0.3 | https://cran.r-project.org |
| Software, algorithm | RStudio | Posit Software | v1.4.1103; RRID:SCR_000432 | https://posit.co/download/rstudio-desktop/ |
| Software, algorithm | BLISAcounts | This paper | N/A | https://github.com/RuiyuRayWang/BLISAcounts/ |

## Reagents and reagent preparation

Detailed sequences and modifications of DNA oligos are listed in *Supplementary file 3*. Detailed sequences of the vectors used in this study are listed in *Supplementary file 4*. DNA oligos were synthesized by Thermo Fisher Scientific, Genewiz, and Sangon Biotech. The sequences of HCR (initiators and amplifiers) are adopted from previous reports (*Choi et al., 2014*; *Wang et al., 2020*). All oligos were dissolved in nuclease-free water (Thermo Fisher Scientific, AM9932) and stored at –20°C. Polypeptides (Gly-Val-Gly-Lys(N$_3$)-Arg-Gly, GVG-K(N$_3$)-RG) with azide modification on lysine residues were synthesized by Scilight-Peptide (Beijing, China). Sodium tetraborate decahydrate (NaB; S9640), Triton X-100 (T8787), dextran sulfate (D8906), TCEP (646547), Tween-20 (P9416), protease inhibitor cocktail (PIC; P8340), and TMB (T0440) were purchased from Sigma-Aldrich. 20× sodium chloride citrate (SSC) buffer (AM9763), sheared Salmon sperm DNA (sssDNA; AM9680), cell extraction buffer (FNN0011), and protein-free blocking buffer (PFBB; 37572) were purchased from Thermo Fisher Scientific. Phosphatase inhibitor cocktail (PPIC, 5870S) was purchased from Cell Signaling Technology. Dulbecco's Modified Eagle Medium (DMEM), Minimum Essential Medium (MEM), RPMI 1640 Medium, and penicillin-streptomycin were purchased from Gibco. FBS was purchased from Gibco and HAKATA (Shanghai, China).

## Protein expression and purification

Recombinant expression of OaAEP1 (C247A) was performed as previously described (*Yang et al., 2017*; *Rehm et al., 2019*; *Harris et al., 2015*). Briefly, pET28a-His$_6$-Ubiquitin-OaAEP1 (C247A) was transformed into *E. coli* SHuffle, and the protein expression was induced overnight with 1 mM IPTG at 18°C. Cells were harvested by centrifugation, resuspended in lysis buffer (50 mM NaH$_2$PO$_4$, pH 8.0, 300 mM NaCl, 10 mM imidazole), and lysed by sonication. Cell debris was cleared via 1 hr centrifugation at 39,000×*g* at 4°C. The supernatant was bound to Ni-NTA resins, washed with AEP wash buffer (50 mM NaH$_2$PO$_4$, pH 8.0, 0.3 M NaCl, 20 mM imidazole), and eluted with elution buffer (50 mM NaH$_2$PO$_4$, pH 8.0, 0.3 M NaCl, 250 mM imidazole). To self-activate OaAEP1, the eluted fraction was diluted in PBS buffer and incubated overnight at room temperature (RT), with 1 mM EDTA and 0.5 mM TCEP added to the solution, and the pH adjusted to 4.0 using glacial acetic acid. After being filtered by a 0.22 μm membrane and 1:8 diluted in 50 mM acetate at pH 4.0, the mature protein was purified further by cation exchange (HiTrap SP column) and eluted with a salt gradient (0–100% of storage buffer [50 mM sodium acetate, pH 4.0, 10% glycerol, 1 M NaCl]; 10 column volumes). Fractions containing mature AEP were pooled and concentrated using a 10 K MWCO concentrator (Sartorius). The final product was analyzed by SDS-PAGE, quantified via A$_{280}$ reading (NanoPhotometer N60, IMPLEN) according to the protein molecular weight and extinctions coefficient, and then stored in aliquots at –80°C until use.

Nanobodies against the Fc domain of IgG (TP897 against rabbit IgG and TP1107 against mouse IgG1) were expressed and purified according to previously published methods (*Pleiner et al., 2018*; *Fabricius et al., 2018*). Briefly, pET21a-TP897-NGL-His$_6$ and pET21a-TP1107-NGL-His$_6$ were

transformed into *E. coli* SHuffle. Nanobody expression was induced for 16–18 hr with 1 mM IPTG at 28°C. After being harvested by centrifugation, cells were resuspended in Nb lysis buffer (50 mM HEPES, pH 7.5, 300 mM NaCl, 5 mM imidazole, 10% glycerol) and lysed by sonication. Cell debris was removed by centrifugation, and the supernatant was applied to Co-NTA resins. The resins were washed with Nb wash buffer (20 mM HEPES, pH 7.5, 300 mM NaCl, 20 mM imidazole, 10% glycerol) and the His-tagged nanobodies were eluted with Nb elution buffer (20 mM HEPES, pH 7.5, 300 mM NaCl, 500 mM imidazole, 10% glycerol). The nanobodies were further purified and buffer exchanged by size exclusion column (SEC) in Nb exchange buffer (20 mM HEPES, pH 7.5, 300 mM NaCl, 10% glycerol). Fractions containing nanobodies were pooled and concentrated using Amicon 3 K MWCO concentrator (Millipore). Protein concentrations were determined by $A_{280}$ reading as described above. Nanobodies were stored in aliquots at –80°C until use.

pET28a-GFP and pET28a-mCherry were transformed into *E. coli* BL21(DE3). Protein expression was induced for 16–18 hr with 0.5 mM IPTG at 16°C. GFP and mCherry were purified using Co-NTA resins as described above and were further purified by SEC in PBS. Proteins were concentrated using Amicon 10 K MWCO concentrator and quantified by $A_{280}$ reading.

## C-terminal labeling of nanobodies by OaAEP1 reaction

Purified NGL-tagged nanobodies (Nb-NGL-His$_6$) were labeled C-terminally using GVG-K(N$_3$)-RG polypeptides by OaAEP1 reaction as previously described (*Rehm et al., 2019*). This was carried out by reacting 50 µM nanobody, 1 mM GVG-K(N$_3$)-RG, and 750 nM OaAEP1 (C247A) in reaction buffer (100 mM sodium phosphate buffer, pH 6.5, supplemented with 2 mM DTT) overnight by gentle shaking at RT. The reaction mixture was then 1:1 diluted in Nb lysis buffer and bound to Ni-NTA resins to remove GL-His$_6$ and unreacted His-tagged nanobodies. Resins were washed with Nb lysis buffer until $A_{280}$ reached baseline. Flow-through and wash fractions containing azide-labeled nanobodies were pooled, and the excess GVG-K(N$_3$)-RG polypeptides were removed using a 3000 MWCO concentrator. The azide-labeled nanobodies (Nb-N$_3$) were quantified by $A_{280}$ reading, analyzed by SDS-PAGE, and then stored at –20°C.

## Nanobody-fluorophore conjugation

Azide-labeled nanobodies (40 µM) were mixed with 10-fold molar excess of DBCO-Cy3 (777366, Sigma-Aldrich) or DBCO-Cy5 (777374, Sigma-Aldrich) and incubated at RT with shaking for 4 hr. Excess fluorophores were removed by a Zeba spin desalting column (7 K MWCO) twice. Protein concentration and the degree of labeling were measured by NanoPhotometer N60 with dye correction.

## Nanobody-DNA oligonucleotide conjugation

For Nb-SS-HCR initiator and Nb-SS-DNA barcode, we used DBCO-PEG$_3$-SS-NHS (10 mM in anhydrous DMSO; CP-2089, Conju-Probe, LLC) as the linker. HCR initiators and DNA barcodes modified with a 5′ NH$_2$-C6 moiety were reacted with DBCO-PEG$_3$-SS-NHS (20-fold molar excess) in 0.091 M NaB (pH 8.5) at RT for 2 hr by gentle shaking. To remove excess linker, oligonucleotides were precipitated by mixing with a one-tenth volume of 3 M sodium acetate (pH 5.2) and two volumes of cold absolute ethanol for 1 hr at –20°C. After a 30 min spin at 20,000×*g* at 4°C, the supernatant was removed, and the pellet was carefully rinsed twice with cold 70% ethanol. The pellet was then air-dried and re-dissolved in TE buffer.

The DBCO-PEG$_3$-SS-conjugated oligos or HCR initiators with a 5′ DBCO moiety (twofold molar excess) were mixed with 25 µM azide-labeled nanobodies and incubated at 4°C with shaking for 16 hr. The DNA oligo-conjugated nanobodies were analyzed by SDS-PAGE and then stored at –20°C.

## DNA barcoding to IgG antibodies by MaMBA strategy

Antibodies were premixed with threefold molar excess of DNA oligo-conjugated nanobodies in 50 µL premixing buffer (0.1% BSA, 0.05% Triton X-100 in PBS) at RT with shaking for 1–3 hr. The mixtures were then added to Amicon Ultra-0.5 100 K centrifugal filters and washed 10 times with 400 µL PBS containing 0.05% Tween-20 and 5 mM EDTA. Alternatively, the mixtures were incubated with normal IgG-conjugated beads (Thermo Fisher, 10500C or 31903) for 10 min at RT, and the flow-through was collected. Purified DNA-barcoded antibodies were stored at 4°C with 0.03% NaN$_3$ or at –20°C with 50% glycerol and used within 1 week.

## Mice

Animal care and use followed the approval of the Animal Care and Use Committee of the National Institute of Biological Sciences, Beijing (Approval ID: NIBSLuoM15C), in accordance with the Regulations for the Administration of Affairs Concerning Experimental Animals of China. C57BL/6N mice were purchased from Beijing Vital River Laboratory Animal Technology Co., Ltd (China), and adult (8–12 weeks of age) mice of either sex were used. Mice were maintained with a 12 hr light/dark photoperiod (light on at 8 AM) and were provided food and water ad libitum.

## Tissue sample preparation for immunohistochemistry

Mice were anesthetized with an overdose of pentobarbital and perfused intracardially with PBS, followed by 4% paraformaldehyde (PFA; 4% [wt/vol] in PBS). Brains were dissected out and postfixed in 4% PFA for 4 hr at RT. Samples were cryoprotected in 30% sucrose until they sank. Coronal sections (35 µm) were prepared on a Cryostat microtome (Leica CM1950).

Human skin samples were obtained surgically from 28-year-old female arm skin with psoriasis and 23-year-old male healthy arm skin. Samples were embedded in O.C.T. compound, frozen at –80°C, and sectioned into 30-µm-thick specimens. Skin sections were fixed in 4% PFA for 15 min and washed three times with PBS.

## Immunostaining by *mis*HCR/*mis*HCR$^n$

Detailed information about antibodies and Nb-HCR initiators (or Nb-SS-HCR initiators) used in this study is listed in Key resources table and *Supplementary file 2*. Samples were permeabilized with 0.3% Triton X-100 in PBS (PBST) and blocked with blocking buffer (2% BSA, 5 mM EDTA, 0.3% Triton X-100 in PBS) for 1 hr at RT. MaMBA-generated Ab-HCR initiators (or Ab-SS-HCR initiators) were pooled and diluted into incubation buffer (1% BSA, 0.1% Triton X-100, 5 mM EDTA, 0.5 mg/mL sssDNA, 1% dextran sulfate, 150 mM NaCl, 0.05% NaN$_3$ in PBS) supplemented with excess corresponding Nb-NGL-His$_6$. Tissue sections were incubated with the Ab-HCR initiators (or Ab-SS-HCR initiators) pool at 4°C for 12–16 hr and then washed three times with washing buffer (2% BSA, 0.1% Triton X-100 in PBS) for 10 min. After being washed twice with PBS for 5 min, samples were postfixed using 5 mM BS(PEG)$_5$ (A35396, Thermo Fisher Scientific) in PBS for 1 hr at RT, followed by quenching in Tris-buffered saline for 10 min. For control immunostaining experiments, tissue sections were incubated with antibodies diluted in blocking buffer at 4°C for 12–16 hr and then washed three times in PBST, followed by (1) incubation with Nb-fluorophore or Nb-HCR initiator in incubation buffer for 2 hr at RT; or (2) incubation with a mixture containing equal amounts of Cy3-conjugated anti-rabbit IgG antibody (111-165-008, Jackson ImmunoResearch) and Nb-HCR initiator (B4 I1) in incubation buffer for 2 hr at RT. Samples were then washed three times with PBST.

Before performing the next round of misHCR, tissue sections with anchored Ab-SS-HCR initiators were incubated with 50 mM TCEP in PBS (to remove HCR initiators via reductive cleavage) for 15 min at RT, followed by three washes with PBST. Samples were incubated in blocking buffer for 1 hr at RT, and a new round of staining was performed as above.

For the experiment for spatial resolution measurement, HeLa cells were fixed in 4% PFA for 10 min, washed three times with PBST, and blocked with a blocking buffer for 30 min at RT. For the standard method (2nd Ab), cells were stained by antibodies against α-tubulin (1:4000 diluted; T5168, Sigma-Aldrich) and Alexa Fluor (AF) 647-conjugated secondary antibodies (1:1000 diluted; 715-605-151, Jackson ImmunoResearch). For *mis*HCR, cells were stained by Ab-HCR initiator (B4 I1), followed by HCR using AF647-conjugated amplifiers.

## HCR amplification

The basic HCR amplification process was common to all the experiments. Samples were blocked in amplification buffer (5× SSC buffer, 0.1% [vol/vol] Tween-20, 10% dextran sulfate) at RT for 1 hr. Each pair of fluorophore-conjugated HCR amplifiers was snap-cooled separately in 5× SSC buffer by being heated at 95°C for 90 s using a PCR machine, immediately cooled on ice for 5 min, and then kept in the dark at RT for over 30 min. The amplifiers were mixed in amplification buffer to a final concentration of 12.5 nM for each amplifier. Samples were incubated with HCR amplifiers overnight at RT and washed three times with 5× SSCT (5× SSC buffer, 0.1% Tween-20) before imaging. The incubation time for HCR can be tuned by adjusting the concentration of HCR amplifiers.

To perform multi-round imaging, samples were washed twice with 50% formamide in 0.1× PBS at 37°C for 5 min to remove the hybridized amplifiers and then washed three times with PBST. After imaging, HCR initiators and hybridized HCR amplifiers were removed by TCEP treatment as described above. Sections were blocked again in the amplification buffer for 1 hr at RT to start the next round of HCR amplification.

## Fluorescence imaging

For image acquisition, samples were mounted on microscope slides in 50% glycerol containing 1 μg/mL DAPI (D9542, Sigma-Aldrich). Confocal microscopy was mainly performed on a Zeiss LSM880 using a ×20/NA-0.8 or ×100/NA-1.45 objective, or on a Leica SP8 using a ×20/NA-0.75 or ×10/NA-0.40 objective. DAPI was visualized using a 405 nm laser; AF488 was visualized using a 488 nm laser; Cy3, AF546, and AF594 were visualized using a 552 nm laser; and Cy5 and AF647 were visualized using a 638 nm laser. The images were 1024×1024 or 512×512 pixels and acquired with a frame averaging of two. Images were processed using Zeiss ZEN, Leica LAS X, and ImageJ (*Schindelin et al., 2012*), and colored for display using ImageJ and Photoshop. For multi-round images, a basic alignment and registration were done by descriptor-based registration plugin (*Preibisch et al., 2010*) based on the DAPI channel of each image using ImageJ.

For 6-color multiplexing of human skin samples by *mis*HCR, the entire experiment was performed with three image rounds (round 1, PDGFRα, KRT14, and DCT; round 2, CD31; round 3, αSMA and CD45). For 9-color multiplexing of mouse brain slices by misHCR$^n$, three stain rounds with one image round of each were performed (first *mis*HCR round, MAP2, DDC, and nNOS; second *mis*HCR round, NeuN, Iba1, and GFAP; third *mis*HCR round, Th, NF-H, and Tph2). For 12-color multiplexing of mouse brain slices by *mis*HCR$^n$, two rounds of *mis*HCR were performed with four image rounds (round 1, nNOS and GFAP; round 2, NeuN, Th and NF-H; round 3, Tph2 and Orexin A; round 4, GABA) and three image rounds (round 1, Tmem119 and DDC; round 2, 5-HT; round 3, Iba1), respectively.

## Cell culture and transfection

HEK293T cells (ATCC CRL-3216) and HeLa cells (ATCC CCL-2) were cultured in DMEM supplemented with 10% (vol/vol) FBS, 100 units/mL penicillin, and 100 units/mL streptomycin at 37°C with 5% $CO_2$. U87 cells (ATCC HTB-14) were cultured in MEM with 10% FBS, 100 units/mL penicillin, and 100 units/mL streptomycin. U937 cells (ATCC CRL-1593.2) were cultured in RPMI 1640 Medium with 10% FBS, 100 units/mL penicillin, and 100 units/mL streptomycin. All cell lines were authenticated by STR profiling and free of mycoplasma contamination.

Cells were seeded in a 24-well plate and grown to 70–90% viable. For immunostaining, cells were cultured on coverslips pretreated with poly-L-lysine solution (P8920, Sigma-Aldrich). Cell medium was replaced by fresh complete medium 2 hr before transfection. For each well, 0.5 μg plasmid and 0.5 μL DNA transfection reagent (Neofect) were mixed in 50 μL serum-free medium and incubated for 15–20 min at RT. The mixture was added to the cells with further incubation for 18–24 hr at 37°C with 5% $CO_2$ before assaying for transgene expression. For cells transfected with P$_{Tight}$-EGFP-PGK-rtTA, Tet system-approved FBS (Biological Industries) was used.

For magnetic bead-based BLISA, cells were seeded in six-well (2–3 ×$10^5$/well) and grown for 24 hr in complete medium. Cells were subjected to an overnight starvation protocol or cultured in the presence of 0.2% FBS condition for a designated duration prior to drug treatments.

## Drug treatment and cell lysis

HEK293T cells transfected with P$_{Tight}$-EGFP-PGK-rtTA were treated with ATc (541642, J&K) at varying concentrations (0–10 μM). HEK293T cells transfected with 3×CRE-mCherry-PEST were treated with FSK (F3917, Sigma-Aldrich) at varying concentrations (0–100 μM) in the presence of 0.3 mM 3-isobutyl-1-methylxanthine (IBMX; I5879, Sigma-Aldrich). HEK293T cells co-transfected with P$_{Tight}$-EGFP-PGK-rtTA and 3×CRE-mCherry-PEST were treated with ATc and FSK at varying concentrations in the presence of 0.3 mM IBMX, according to the protocol as described in the figure panel. After incubation for 6–7 hr at 37°C with 5% $CO_2$, cells were lysed in cell extraction buffer supplemented with 1 mM PMSF and 1 × PIC following the manufacturer's instructions, quantified using BCA protein assay (23225, Thermo Fisher Scientific) and stored in aliquots at –80°C until use.

U87 and U937 cells seeded in six-well plates were treated with 10% FBS for 10 min, 100 mM anisomycin (CS-4981, MedChem Express) for 1 hr, or 20 mM $H_2O_2$ for 10 min, respectively. Then, cells were lysed in denaturing cell lysis buffer (6 M urea, 0.5% Triton X-100, 1 mM EDTA, 5 mM NaF in PBS, pH 7.2–7.4) supplemented with 1 × PIC and 2 × PPIC. After being lysed on ice for 15 min, samples were centrifuged at 2000×$g$ for 5 min, and the supernatants were assayed immediately or stored at –80°C until use. Cell lysates from three wells were pooled for each group. Before use, sample protein concentrations were measured using a BCA protein assay and equilibrated with the denaturing cell lysis buffer. Cell lysates were diluted sixfold with dilution buffer (0.5% Triton X-100, 1 mM EDTA, 5 mM NaF in PBS, pH 7.2–7.4).

## Serum sample collection

Human serum samples were collected from volunteers of laboratory workers and from individuals who undertook routine physical examinations for health screening in hospitals, approved by the Human Research Ethics Committee of the Chinese Institute for Brain Research, Beijing. No informed consent or consent to publish was required for this study, as no human participants were involved. Only anonymized human serum samples were used, with no access to donor information.

Serum samples for anti-RBD IgG detection were collected in October 2021 from a cohort of 517 donors. The majority of these samples (n=493) were collected from healthy individuals undergoing routine physical examinations during the initial phase of the SARS-CoV-2 vaccination campaign in Beijing, China. Due to privacy protection regulations, information regarding vaccination status and SARS-CoV-2 infection history (including infection status, vaccination status, vaccine type and manufacturer, number of doses received, and dates of administration) was not collected for these donors. To better validate the performance of BLISA, we obtained serum samples from additional 24 donors who voluntarily disclosed their SARS-CoV-2 vaccination records. Of these, 23 had received at least one dose of an inactivated SARS-CoV-2 vaccine, while one donor had neither been vaccinated nor tested positive for SARS-CoV-2 (the negative control).

Serum samples for HBV antigen detection were collected in June 2021 (cohort 1) and October 2023 (cohort 2) from donors undergoing routine physical examinations. Cohort 1 was collected from 529 donors, including 6 positive (HBsAg$^+$HBeAg$^+$) and 25 negative (HBsAg$^-$HBeAg$^-$) donors. Cohort 2 was collected from 600 donors, including 100 samples from donors that had been clinically diagnosed as HBV positive. Duplicate ELISAs were conducted for 115 samples in cohort 1 and all samples in cohort 2. Only samples demonstrating reproducible results across both replicates were included in subsequent analyses.

## BLISA on microplate surface

Detailed information about antibodies and the Nb-SS-DNA barcodes used is listed in *Supplementary file 2*. HRP-conjugated secondary antibodies against rabbit IgG (1:30,000 diluted; 31460, Thermo Fisher Scientific) and against mouse IgG (1:30,000 diluted; 31430, Thermo Fisher Scientific) were used.

For cell lysate samples spiked in with purified GFP and mCherry, 50 µL purified proteins at a designated concentration were mixed with 1 µg/mL wild-type HEK293T cell lysate in coating buffer (50 mM carbonate buffer, pH 9.6). The mixtures were immobilized on a high-binding polystyrene microplate (449824, Thermo Fisher Scientific) for 2 hr at 37°C. Coated microplates were washed three times with plate wash buffer (5 mM EDTA, 0.05% Tween-20 in PBS) and then blocked for 1 hr at RT in 200 µL blocking buffer (0.05% Tween-20 in PFBB), followed by three additional wash steps using plate wash buffer. MaMBA-generated Ab-SS-DNA barcodes were pooled into incubation solution (blocking buffer supplemented with 5 mM EDTA, 0.4 mg/mL sheared Salmon sperm DNA, 2 µM TP1107-NGL-His$_6$, and 2 µM TP897-NGL-His$_6$). Samples were then incubated with the Ab-SS-DNA barcodes pool for 1 hr at 37°C. Microplates were washed five times with plate wash buffer and three times with PBS containing 5 mM EDTA. 50 µL elution buffer (20 mM TCEP in PBS) was added to each well and incubated for 5–10 min at RT. Elution buffer containing retrieved DNAs was assayed immediately or stored at –20°C until analysis. 1 µL of elution buffer containing retrieved DNAs was used as a template for qPCR or sequencing library preparation. For ELISA experiments, samples were incubated with antibodies (diluted in PFBB) for 1 hr at 37°C and washed three times with plate wash buffer, followed by incubation with HRP-conjugated secondary antibodies diluted in PFBB for 1 hr at 37°C and washing five times with plate wash buffer. HRP-driven colorimetric readout was done by adding 50 µL TMB solution

and incubating for 10 min at RT. 50 µL 1.8 N $H_2SO_4$ was subsequently added to stop the reaction. Absorbance at 450 nm was measured immediately by microplate reader (Spark, Tecan). Values were corrected by subtracting the absorbance of blank wells. For cell lysate samples containing endogenously expressed GFP and mCherry, cell lysates were diluted to 1 µg/mL using coating buffer and immobilized on the microplate. Subsequent BLISAs were conducted as described above.

For anti-RBD IgG detection in human serum samples, SARS-CoV-2 Spike RBD-His recombinant proteins (40592-V08B, Sino Biological) (1 µg/mL diluted in PBS) were immobilized on the microplate at RT overnight and washed three times with plate wash buffer using microplate washer (Wellwash Versa Microplate Washer, Thermo Fisher Scientific). After incubation in blocking buffer for 1 hr at RT and three times of plate washing, a mixture of 1 µL serum sample and 49 µL PBS was added to the coated microplate and incubated for 1 hr at 37°C. After five times of plate washing, captured anti-RBD IgGs were detected using Ab-SS-DNA barcodes (antibodies against human IgG in incubation buffer) for 1 hr at 37°C. The plates were washed five times with plate wash buffer and three times with PBS containing 5 mM EDTA. DNA barcodes were retrieved using 50 µL elution buffer containing 10 pM normalization DNA oligos and incubated for 5–10 min at RT.

For HBsAg and HBeAg detection in human serum samples, 40 µL serum sample, 10 µL PBS, and 50 µL Ab-SS-DNA barcodes pool were mixed and added to microplates pre-coated with capture antibodies and incubated for 1 hr at 37°C, followed by washing and DNA retrieval with 1 pM normalization DNA oligos.

## BLISA on magnetic beads

Detailed information about the antibody pairs and the DNA barcodes is listed in *Supplementary file 2*. 18 mg epoxy magnetic beads (Beijing Yunci Technology) were mixed with 0.1 M sodium phosphate (pH 7.4), capture antibodies (72-288 mg, corresponding to one vial of antibodies in the kit), and 1 M ammonium sulfate for 16–24 hr at 37°C. Magnetic beads were then washed three times with PBS containing 1% Triton X-100, followed by blocking in PBS with 1% BSA and 0.05% Tween-20 for 2 hr at RT. After magnetic isolation, antibody-coupled magnetic beads (10 mg/mL) were maintained in PBS with 0.1% BSA, 0.05% Tween-20, and 0.03% $NaN_3$ at 4°C. For multiplexing experiments, antibody-coupled beads for each target were mixed before use.

After washing twice with beads wash buffer (0.1% Triton X-100 in PBS), antibody-coupled magnetic beads were resuspended in wash buffer and aliquoted into a 96-well deep well plate (501102, NEST). Samples were then added and mixed with antibody-coupled magnetic beads for 30 min at RT using a plate mixer (800 rpm). Magnetic beads were isolated using a 96-well magnetic stand, followed by 3× wash steps with beads wash buffer. Antigen-captured beads were incubated with Ab-SS-DNA barcodes pool in beads incubation buffer (0.5% BSA, 0.05% Triton X-100, 0.1% dextran sulfate, 0.4 mg/mL sheared Salmon sperm DNA, 5 µM TP1107- NGL-His$_6$, 2 µM TP897-NGL-His$_6$ in PBS) for 30 min at RT using a plate mixer (800 rpm), followed by 5× wash steps with beads wash buffer. Beads were rinsed twice with PBS, and DNA barcodes were retrieved using 50 µL elution buffer (20 mM TCEP in PBS) for 5–10 min at RT using a plate mixer (1000 rpm). After magnetic isolation, solutions containing retrieved DNAs were transferred to new tubes and measured immediately or stored at –20°C until analysis.

## Quantitative PCR

One microliter of solutions containing retrieved DNAs was used for qPCR (triplicates for each sample; 20 µL reaction volume, SYBR qPCR master mix, Q712, Vazyme) using a CFX 96 machine (Bio-Rad). Primer sequences are listed in *Supplementary file 5*.

## Sequencing library preparation

For the two-step PCR protocol (*Figure 4—figure supplement 2*), a 25 µL PCR to add well-specific barcode was performed for each sample using 1 µL retrieved DNA barcodes, 0.1 mM dNTPs, 0.5 U Q5 high-fidelity DNA polymerases (NEB), 1×Q5 reaction buffer, 0.4 µM well-specific forward primers, and 0.4 µM reverse primers (*Supplementary file 6*). Retrieved DNA barcodes were amplified over nine cycles of 98°C for 10 s, 50°C for 30 s, and 72°C for 10 s. For each microplate, 1 µL PCR product from each well was pooled and subsequently purified using 1.8× AMPure XP beads (Beckman Coulter), followed by elution using 30 µL nuclease-free water. An additional 50 µL PCR for each microplate to

add sequencing adaptors was performed using 29 μL purified pooled PCR products, 0.2 mM dNTPs, 1 U Q5 polymerase, 1×Q5 reaction buffer, 0.5 μM adaptor forward primer, and 0.5 μM adaptor reverse primer containing i7 index. Well-barcoded DNAs were amplified over nine cycles of 98°C for 10 s, 50°C for 30 s, and 72°C for 10 s. PCR products were purified using 1.8×AMPure XP beads. Final sequencing libraries were eluted in 30 μL nuclease-free water.

For the three-step PCR protocol with phasing spacers addition (*Figure 5—figure supplement 2*), a 25 μL PCR to add well-specific barcode was performed for each sample using 1 μL retrieved DNA barcodes, 0.1 mM dNTPs, 0.5 U Q5 polymerases, 1×Q5 reaction buffer, 0.4 μM well-specific forward primers, and 0.4 μM reverse primers (*Supplementary file 6*). Retrieved DNA barcodes were amplified over nine cycles of 98°C for 10 s, 50°C for 30 s, and 72°C for 10 s. For each microplate, 0.73 μL PCR product from each well was pooled and purified using 1.8×AMPure XP beads, followed by elution using 85 μL nuclease-free water. Eight 25 μL PCRs were performed to add phasing spacers using 10 μL purified pooled PCR products, 0.1 mM dNTPs, 0.5 U Q5 polymerases, 1×Q5 reaction buffer, 0.4 μM PS forward primers, and 0.4 μM PS reverse primers (*Supplementary file 6*). Well-barcoded DNAs were amplified over nine cycles of 98°C for 10 s, 55°C for 30 s, and 72°C for 10 s. PCR products (8.75 μL of each reaction) were pooled and purified using 1.8×AMPure XP beads, followed by elution using 30 μL nuclease-free water. A final 50 μL PCR was performed to add sequencing adaptors using 5 μL phasing spacer added DNAs, 0.2 mM dNTPs, 0.5 U Q5 polymerases, 1×Q5 reaction buffer, 0.5 μM forward primers, and 0.5 μM reverse primers containing i7 index (plate index). NGS libraries were amplified over four cycles of 98°C for 10 s, 60°C for 30 s, and 72°C for 10 s. NGS libraries were then purified using 1.8×AMPure XP beads and eluted with 20 μL nuclease-free water. NGS libraries for each microplate were individually analyzed and quantified before pooling. NGS was performed on a HiSeq 2500 System (Illumina).

## NGS data analysis

We devised a Snakemake (*Mölder et al., 2021*) pipeline to extract and quantify well barcode (well-bc), antibody-specific barcode (Ab-bc), and UMI from the BLISA libraries.

For standard BLISA libraries, whitelists for well-bc, Ab-bc, and UMIs were generated at positions 1–6, 21–26, and 27–41, respectively, from the sequencing reads, using the 'whitelist' command from UMI-tools (*Smith et al., 2017*). The whitelist was filtered with respect to a ground truth list, and the well-bc, Ab-bc, and UMIs were extracted using the 'extract' command. For BLISA libraries with staggered sequences, two anchor sequences 'GACAAGTGGCCACAAACCACCAG' and 'CTTGTGGA AAGGACGAAACA' located between the well-bc, Ab-bc, and UMIs were used to identify and extract the barcodes and UMIs using Biopython (*Cock et al., 2009*).

Base pair corrections were performed on barcode-UMI combinations 2 Hamming distance from the ground truth barcode-UMI combinations as initially designed in the experiments.

The extracted barcodes and UMIs were demultiplexed and aggregated into dataframes. Analyses and visualization were performed using tidyverse (*Wickham et al., 2019*).

## Statistics and reproducibility

Statistical significance was evaluated using Prism 9 (GraphPad). All fluorescence intensity, immunostaining, BLISA qPCR, western blot, and ELISA experiments were independently performed ≥3 times with similar results. HTS-based BLISA experiments were performed once.

## Acknowledgements

We thank Dr. Ting Chen (NIBS, Beijing, China) for her support in the experiments on human skin samples. The reagents for anti-RBD IgG and HBV antigens detection experiments were supported by Beijing Wantai Biological Pharmacy, Peking University International Hospital, and Beijing Tiantan Hospital. We thank all members of RL's lab and ML's lab for their assistance in this study. RL is supported by Beijing Nova Program (20230484303), National Natural Science Foundation of China (32322032), China Brain Initiative Grant (2022ZD0206700), and the Beijing Municipal Government. ML is supported by China Brain Initiative Grant (STI2030-Major Projects 2021ZD0202803), the Research Unit of Medical Neurobiology at Chinese Academy of Medical Sciences (2019RU003), New Cornerstone Investigator Program, and the Beijing Municipal Government. The funders had no role in the study design, data collection, and analysis, manuscript preparation, or decision to publish the manuscript.

# Additional information

## Competing interests

Shilin Zhong, Rui Lin, Minmin Luo: The National Institute of Biological Sciences (NIBS), Beijing, China has filed patent applications related to this work with RL, SZ, and ML listed as inventors (PCT/CN2022/106344). The other authors declare that no competing interests exist.

## Funding

| Funder | Grant reference number | Author |
|---|---|---|
| National Natural Science Foundation of China | 32322032 | Rui Lin |
| China Brain Initiative | 2022ZD0206700 | Rui Lin |
| Beijing Nova Program | 20230484303 | Rui Lin |
| China Brain Initiative | 2021ZD0202803 | Minmin Luo |
| Chinese Academy of Medical Sciences | Research Unit of Medical Neurobiology 2019RU003 | Minmin Luo |
| New Cornerstone Investigator Program | | Minmin Luo |

The funders had no role in study design, data collection and interpretation, or the decision to submit the work for publication.

## Author contributions

Shilin Zhong, Conceptualization, Data curation, Formal analysis, Validation, Investigation, Visualization, Methodology, Writing – original draft, Writing – review and editing; Ruiyu Wang, Formal analysis, Writing – original draft; Xinwei Gao, Formal analysis; Qingchun Guo, Supervision; Rui Lin, Conceptualization, Supervision, Investigation, Writing – original draft, Writing – review and editing; Minmin Luo, Conceptualization, Resources, Supervision, Funding acquisition, Writing – original draft, Project administration, Writing – review and editing

## Author ORCIDs

Shilin Zhong ⓘ https://orcid.org/0000-0002-0295-1294
Ruiyu Wang ⓘ https://orcid.org/0000-0002-7133-7693
Rui Lin ⓘ https://orcid.org/0000-0003-1044-883X
Minmin Luo ⓘ https://orcid.org/0000-0003-3535-6624

## Ethics

Human subjects: No informed consent or consent to publish was required for this study, as no human participants were involved. Only anonymized human serum samples were used, with no access to donor information. Human serum samples were collected from volunteers of laboratory workers and from individuals who undertook routine physical examinations for health screening in hospitals, approved by the Human Research Ethics Committee of the Chinese Institute for Brain Research, Beijing.
Animal care and use followed the approval of the Animal Care and Use Committee of the National Institute of Biological Sciences, Beijing (Approval ID: NIBSLuoM15C), in accordance with the Regulations for the Administration of Affairs Concerning Experimental Animals of China.

Reviewer #1 (Public review): https://doi.org/10.7554/eLife.105225.3.sa1
Author response https://doi.org/10.7554/eLife.105225.3.sa2

# Additional files

## Supplementary files

Supplementary file 1. Detailed compound information in HTS-based barcode-linked immunosorbent assay (BLISA).

Supplementary file 2. Antibodies and Nb-DNA oligos (or Nb-SS-DNA oligos).

Supplementary file 3. Detailed sequences and modifications of DNA oligos.

Supplementary file 4. Amino acid sequences of the constructs used in this study.

Supplementary file 5. Primers for quantitative PCR (qPCR)-based barcode-linked immunosorbent assay (BLISA).

Supplementary file 6. Primers for next-generation sequencing (NGS).

MDAR checklist

Source data 1. Statistical source data, related to *Figure 1—figure supplement 1*, *Figure 1—figure supplement 3*, *Figure 1—figure supplement 4*, *Figure 3*, *Figure 3—figure supplement 1*, *Figure 4*, *Figure 4—figure supplements 1 and 2*, *Figure 5*, *Figure 5—figure supplements 1 and 3*, *Figure 6*, *Figure 6—figure supplement 1*.

Source data 2. Unique molecular identifier (UMI) counts of next-generation sequencing (NGS) data, related to *Figure 4—figure supplement 2*, *Figure 5*, *Figure 5—figure supplements 1 and 3*, *Figure 6*, *Figure 6—figure supplements 1 and 2*.

## Data availability

The raw sequence data reported in this paper have been deposited in the Genome Sequence Archive (*Chen et al., 2021*) in National Genomics Data Center (*Xue et al., 2022*), China National Center for Bioinformation / Beijing Institute of Genomics, Chinese Academy of Sciences under the accession GSA: CRA012054, that are publicly accessible at https://ngdc.cncb.ac.cn/gsa/browse/CRA012054. The source image files are provided. Due to the large size, the source image file related to *Figure 2—figure supplement 2* has been deposited on Zenodo (https://doi.org/10.5281/zenodo.15790283). Statistical source data are provided (*Source data 1* and *Source data 2*). Published or publicly available software and algorithms are cited with their version numbers in the Key Resources Table. The code for BLISA analysis is available at https://github.com/RuiyuRayWang/BLISAcounts (copy archived at *Wang, 2024*).

The following datasets were generated:

| Author(s) | Year | Dataset title | Dataset URL | Database and Identifier |
|---|---|---|---|---|
| Shilin Z, Ruiyu W, Xinwei G, Qingchun G, Rui L, Minmin L | 2025 | Modular DNA Barcoding of Nanobodies Enables Multiplexed in situ Protein Imaging and High-throughput Biomolecule Detection | https://ngdc.cncb.ac.cn/gsa/browse/CRA012054 | Genome Sequence Archive, CRA012054 |
| Shilin Z | 2025 | Figure 2–figure supplement 2–Source Data 1 | https://doi.org/10.5281/zenodo.15790283 | Zenodo, 10.5281/zenodo.15790283 |

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
