## [Editor Report · eLife Assessment]

This **fundamental** manuscript presents a practical modification of the orthogonal hybridization chain reaction (HCR) technique, a promising yet underutilized method with broad potential for future applications across various fields. The authors advance this technique by integrating peptide ligation technology and nanobody-based antibody mimetics-cost-effective and scalable alternatives to conventional antibodies-into a DNA-immunoassay framework that merges oligonucleotide-based detection with immunoassay methodologies. Notably, they demonstrate with **compelling** evidence that this approach facilitates a modified ELISA platform capable of simultaneously quantifying multiple target protein expression levels within a single protein mixture sample.

---

## [Referee Report · Reviewer #1 (Public review)]

Summary:

This fundamental study presents a practical modification of the orthogonal hybridization chain reaction (HCR) technique, a promising yet underutilized method with broad potential for future applications across various fields. The authors advance this technique by integrating peptide ligation technology and nanobody-based antibody mimetics-cost-effective and scalable alternatives to conventional antibodies-into a DNA-immunoassay framework that merges oligonucleotide-based detection with immunoassay methodologies. They demonstrate this with compelling evidence that this approach facilitates a modified ELISA platform capable of simultaneously quantifying multiple target protein expression levels within a single protein mixture sample.

Strengths:

The hybridization chain reaction (HCR) technique was initially developed to enable the simultaneous detection of multiple mRNA expression levels within the same tissue. This method has since evolved into immuno-HCR, which extends its application to protein detection by utilizing antibodies. A key requirement of immuno-HCR is the coupling of oligonucleotides to antibodies, a process that can be challenging due to the inherent difficulties in expressing and purifying conventional antibodies.

In this study, the authors present an innovative approach that circumvents these limitations by employing nanobody-based antibody mimetics, which recognize antibodies, instead of directly coupling oligonucleotides to conventional antibodies. This strategy facilitates oligonucleotide conjugation-designed to target the initiator hairpin oligonucleotide of HCR-through peptide ligation and click chemistry.

Weaknesses:

The sandwich-format technique presented in this study, which employs a nanobody that recognizes primary IgG antibodies, may have limited scalability compared to existing methods that directly couple oligonucleotides to primary antibodies. This limitation arises because the C-region types of primary antibodies are relatively restricted, meaning that the use of nanobody-based detection may constrain the number of target proteins that can be analyzed simultaneously. In contrast, the conventional approach of directly conjugating oligonucleotides to primary antibodies allows for a broader range of protein targets to be analyzed in parallel.

Additionally, in the context of HCR-based protein detection, the number of proteins that can be analyzed simultaneously is inherently constrained by fluorescence wavelength overlap in microscopy, which limits its multiplexing capability. By comparison, direct coupling oligonucleotides to primary antibodies can facilitate the simultaneous measurement of a significantly greater number of protein targets than the sandwich-based nanobody approach in the barcode-ELISA/NGS-based technique.

Comments on revisions:

The previous suggestions were well incorporated in the revised manuscript.

---

## [Author Response]

The following is the authors’ response to the original reviews

**Reviewer #1 (Public review):**
Summary:This manuscript presents a practical modification of the orthogonal hybridization chain reaction (HCR) technique, a promising yet underutilized method with broad potential for future applications across various fields. The authors advance this technique by integrating peptide ligation technology and nanobody-based antibody mimetics - cost-effective and scalable alternatives to conventional antibodies - into a DNA-immunoassay framework that merges oligonucleotide-based detection with immunoassay methodologies. Notably, they demonstrate that this approach facilitates a modified ELISA platform capable of simultaneously quantifying multiple target protein expression levels within a single protein mixture sample.Strengths:The hybridization chain reaction (HCR) technique was initially developed to enable the simultaneous detection of multiple mRNA expression levels within the same tissue. This method has since evolved into immuno-HCR, which extends its application to protein detection by utilizing antibodies. A key requirement of immuno-HCR is the coupling of oligonucleotides to antibodies, a process that can be challenging due to the inherent difficulties in expressing and purifying conventional antibodies.In this study, the authors present an innovative approach that circumvents these limitations by employing nanobody-based antibody mimetics, which recognize antibodies, instead of directly coupling oligonucleotides to conventional antibodies. This strategy facilitates oligonucleotide conjugation - designed to target the initiator hairpin oligonucleotide of HCR -through peptide ligation and click chemistry.Weaknesses:The sandwich-format technique presented in this study, which employs a nanobody that recognizes primary IgG antibodies, may have limited scalability compared to existing methods that directly couple oligonucleotides to primary antibodies. This limitation arises because the C-region types of primary antibodies are relatively restricted, meaning that the use of nanobody-based detection may constrain the number of target proteins that can be analyzed simultaneously. In contrast, the conventional approach of directly conjugating oligonucleotides to primary antibodies allows for a broader range of protein targets to be analyzed in parallel.

We would like to clarify that MaMBA was specifically designed to address and overcome the limitations imposed by relying on primary antibodies’ Fc types for multiplexing. MaMBA utilizes DNA oligo-conjugated nanobodies that selectively and monovalently bind to the Fc region of IgG. This key feature allows us to barcode primary IgGs targeting different antigens independently. These barcoded IgGs can then be pooled together after barcoding, effectively minimizing the potential for cross-reactivity or crossover. Therefore, IgGs barcoded using MaMBA are functionally equivalent to those barcoded via conventional direct conjugation approaches with respect to multiplexing capability.

Additionally, in the context of HCR-based protein detection, the number of proteins that can be analyzed simultaneously is inherently constrained by fluorescence wavelength overlap in microscopy, which limits its multiplexing capability. By comparison, direct coupling of oligonucleotides to primary antibodies can facilitate the simultaneous measurement of a significantly greater number of protein targets than the sandwich-based nanobody approach in the barcode-ELISA/NGS-based technique.

As we have responded above, MaMBA barcoding of primary IgGs that target various antigens can be conducted separately. Once barcoded, these IgGs can then be combined into a single pool. Therefore, for BLISA (*i.e.*, the barcode-ELISA/NGS-based technique), IgGs barcoded through MaMBA offer the same multiplexing capability as those barcoded using traditional direct conjugation methods.

In in situ protein imaging, spectral overlap can indeed limit the throughput of multiplexed HCR fluorescent imaging. There are two strategies to address this challenge. As demonstrated in this work with _mis_HCR and _mis_HCRn, removing the HCR amplifiers allows for multiplexed detection using a limited number of fluorescence wavelengths. This is achieved through sequential rounds of HCR amplification and imaging. Alternatively, recent computational approaches offer promising solutions for “one-shot” multiplexed imaging. These include combinatorial multiplexing (PMID: 40133518) and spectral unmixing (PMID: 35513404), which can be applied to _mis_HCR to deconvolute overlapping spectra and increase multiplexing capacity in a single imaging acquisition.

**Reviewer #1 (Recommendations for the authors):**
(1) The introduction of nanobody and peptide ligation technology is a key highlight of this study. To strengthen the manuscript, the authors should provide a more detailed discussion of the principles and applications of HCR in the Introduction or Discussion sections.

We have added a brief discussion of the HCR reaction to the revised manuscript.

(2) It would also be beneficial to include results and/or discussion on how the affinity of nanobody binding to IgG influences the success and accuracy of the technique.

We have added a brief discussion of the IgG nanobodies we used in MaMBA to the revised manuscript.

(3) Additionally, a more detailed explanation of the recognition specificity of the AEP peptide ligase used in this study should be included in the Discussion section. Prior studies have reported on the specificity of amino acid residues positioned at the C-terminus of target A (-5 to -1) and the N-terminus of target B (1 to 3) in AEP-mediated ligation, and integrating this context would enhance clarity.

We have added a brief discussion of the AEP-mediated ligation to the revised manuscript.